# Geometry-aware Instance-reweighted Adversarial Training

**Jingfeng Zhang**[1,2] **Jianing Zhu**[3] **Gang Niu**[1] **Bo Han**[3,1]
**Masashi Sugiyama**[1,4] **Mohan Kankanhalli**[2]

[1]RIKEN Center for Advanced Intelligence Project, Tokyo, Japan
[2]National University of Singapore, Singapore
[3]Hong Kong Baptist University, Hong Kong SAR, China
[4]The University of Tokyo, Tokyo, Japan

```
jingfeng.zhang@riken.jp, csjnzhu@comp.hkbu.edu.hk
gang.niu@riken.jp, bhanml@comp.hkbu.edu.hk
sugi@k.u-tokyo.ac.jp, mohan@comp.nus.edu.sg
```

## Abstract

In *adversarial machine learning*, there was a common belief that *robustness and accuracy hurt each other*. The belief was challenged by recent studies where we can maintain the robustness and improve the accuracy. However, the other direction, we can keep the accuracy and improve the robustness, is conceptually and practically more interesting, since robust accuracy should be lower than standard accuracy for any model. In this paper, we show this direction is also promising. Firstly, we find even *over-parameterized deep networks* may still have *insufficient model capacity*, because adversarial training has an overwhelming *smoothing effect*. Secondly, given limited model capacity, we argue adversarial data should have *unequal importance*: geometrically speaking, a natural data point closer to/farther from the class boundary is less/more robust, and the corresponding adversarial data point should be assigned with larger/smaller weight. Finally, to implement the idea, we propose *geometry-aware instance-reweighted adversarial training*, where the weights are based on *how difficult it is to attack a natural data point*. Experiments show that our proposal boosts the robustness of standard adversarial training; combining two directions, we improve both robustness and accuracy of standard adversarial training.

## 1 Introduction

Crafted *adversarial data* can easily fool the standard-trained deep models by adding human-imperceptible noise to the natural data, which leads to the security issue in applications such as medicine, finance, and autonomous driving (Szegedy et al., 2014; Nguyen et al., 2015). To mitigate this issue, many *adversarial training* methods employ the *most adversarial data* maximizing the loss for updating the current model such as standard adversarial training (AT) (Madry et al., 2018), TRADES (Zhang et al., 2019), robust self-training (RST) (Carmon et al., 2019), and MART (Wang et al., 2020b). The adversarial training methods seek to train an adversarially robust deep model whose predictions are locally invariant to a small neighborhood of its inputs (Papernot et al., 2016). By leveraging adversarial data to smooth the small neighborhood, the adversarial training methods acquire *adversarial robustness* against adversarial data but often lead to the undesirable degradation of *standard accuracy* on natural data (Madry et al., 2018; Zhang et al., 2019).

Thus, there have been debates on whether there exists a trade-off between robustness and accuracy. For example, some argued an inevitable trade-off: Tsipras et al. (2019) showed fundamentally different representations learned by a standard-trained model and an adversarial-trained model; Zhang et al. (2019) and Wang et al. (2020a) proposed adversarial training methods that can trade off standard accuracy for adversarial robustness. On the other hand, some argued that there is no such the trade-off: Raghunathan et al. (2020) showed infinite data could eliminate this trade-off; Yang et al. (2020) showed benchmark image datasets are class-separated.

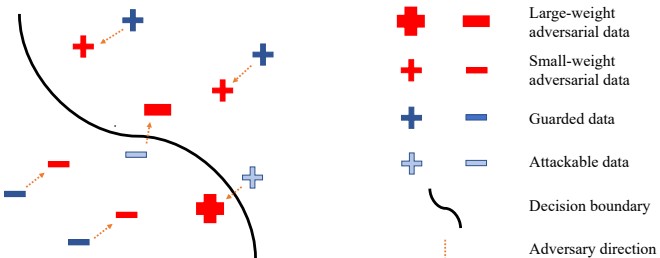

Figure 1: The illustration of GAIRAT. GAIRAT explicitly gives larger weights on the losses of adversarial data (larger red), whose natural counterparts are closer to the decision boundary (lighter blue). GAIRAT explicitly gives smaller weights on the losses of adversarial data (smaller red), whose natural counterparts are farther away from the decision boundary (darker blue). The examples of two toy datasets and the CIFAR-10 dataset refer to Figure 3.

Recently, emerging adversarial training methods have empirically challenged this trade-off. For example, Zhang et al. (2020b) proposed the *friendly adversarial training* method (FAT), employing *friendly adversarial data* minimizing the loss given that some wrongly-predicted adversarial data have been found. Yang et al. (2020) introduced dropout (Srivastava et al., 2014) into existing AT, RST, and TRADES methods. Both methods can improve the accuracy while maintaining the robustness. However, the other direction—whether we can improve the robustness while keeping the accuracy—remains unsolved and is more interesting.

In this paper, we show this direction is also achievable. Firstly, we show over-parameterized deep networks may still have insufficient *model capacity*, because adversarial training has an overwhelming smoothing effect. Fitting adversarial data is demanding for a tremendous model capacity: It requires a large number of trainable parameters or long-enough training epochs to reach near-zero error on the adversarial training data (see Figure 2). The over-parameterized models that fit natural data entirely in the *standard training* (Zhang et al., 2017) are still far from enough for fitting adversarial data. Compared with standard training fitting the natural data points, adversarial training smooths the neighborhoods of natural data, so that adversarial data consume significantly more model capacity than natural data. Thus, adversarial training methods should carefully utilize the limited model capacity to fit the neighborhoods of the important data that aid to fine-tune the decision boundary. Therefore, it may be unwise to give equal weights to all adversarial data.

Secondly, data along with their adversarial variants are not equally important. Some data are geometrically far away from the class boundary. They are relatively *guarded*. Their adversarial variants are hard to be misclassified. On the other hand, some data are close to the class boundary. They are relatively *attackable*. Their adversarial variants are easily misclassified (see Figure 3). As the adversarial training progresses, the adversarially robust model engenders an increasing number of guarded training data and a decreasing number of attackable training data. Given limited model capacity, treating all data equally may cause the vast number of adversarial variants of the guarded data to overwhelm the model, leading to the undesirable robust overfitting (Rice et al., 2020). Thus, it may be pessimistic to treat all data equally in adversarial training.

To ameliorate this pessimism, we propose a heuristic method, i.e., *geometry-aware instance-reweighted adversarial training* (GAIRAT). As shown in Figure 1, GAIRAT treats data differently. Specifically, for updating the current model, GAIRAT gives larger/smaller weight to the loss of an adversarial variant of attackable/guarded data point which is more/less important in fine-tuning the decision boundary. An attackable/guarded data point has a small/large *geometric distance*, i.e., its distance from the decision boundary. We approximate its geometric distance by the least number of iterations $\kappa$ that projected gradient descent method (Madry et al., 2018) requires to generate a misclassified adversarial variant (see the details in Section 3.3). GAIRAT explicitly assigns instance-dependent weight to the loss of its adversarial variant based on the least iteration number $\kappa$.

Our contributions are as follows. (a) In adversarial training, we identify the pessimism in treating all data equally, which is due to the insufficient model capacity and the unequal nature of different data (in Section 3.1). (b) We propose a new adversarial training method, i.e., GAIRAT (its learning objective in Section 3.2 and its realization in Section 3.3). GAIRAT is a general method: Besides standard AT (Madry et al., 2018), the existing adversarial training methods such as FAT (Zhang et al., 2020b) and TRADES (Zhang et al., 2019) can be modified to GAIR-FAT and GAIR-TRADES (in Appendices B.1 and B.2, respectively). (c) Empirically, our GAIRAT can relieve the issue of robust

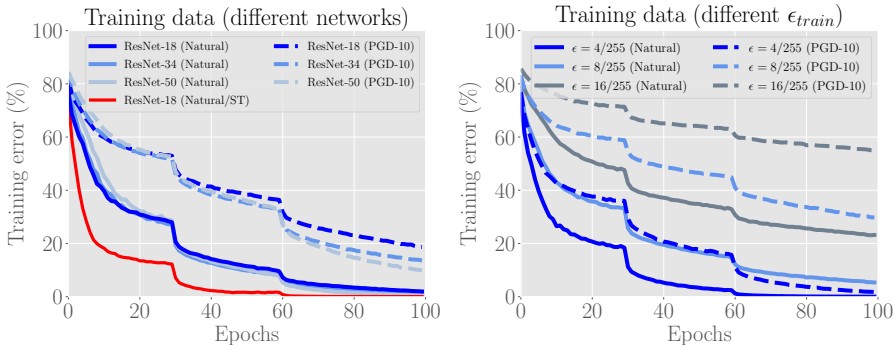

Figure 2: We plot standard training error (Natural) and adversarial training error (PGD-10) over the training epochs of the standard AT on CIFAR-10 dataset. **Left panel**: AT on different sizes of network. The red line represents standard test accuracy by standard training (ST). **Right panel**: AT on ResNet-18 under different perturbation bounds $\epsilon_{\text{train}}$.

overfitting (Rice et al., 2020), meanwhile leading to the improved robustness with zero or little degradation of accuracy (in Section 4.1 and Appendix C.1). Besides, we use Wide ResNets (Zagoruyko & Komodakis, 2016) to corroborate the efficacy of our geometry-aware instance-reweighted methods: Our GAIRAT significantly boosts the robustness of standard AT; combined with FAT, our GAIR-FAT improves both the robustness and accuracy of standard AT (in Section 4.2). Consequently, we conjecture no inevitable trade-off between robustness and accuracy.

## 2 ADVERSARIAL TRAINING

In this section, we review adversarial training methods (Madry et al., 2018; Zhang et al., 2020b).

### 2.1 LEARNING OBJECTIVE

Let $(\mathcal{X}, d_\infty)$ denote the input feature space $\mathcal{X}$ with the infinity distance metric $d_{\text{inf}}(x, x') = \|x - x'\|_\infty$, and $\mathcal{B}_\epsilon[x] = \{x' \in \mathcal{X} \mid d_{\text{inf}}(x, x') \leq \epsilon\}$ be the closed ball of radius $\epsilon > 0$ centered at $x$ in $\mathcal{X}$. Dataset $S = \{(x_i, y_i)\}_{i=1}^n$, where $x_i \in \mathcal{X}$ and $y_i \in \mathcal{Y} = \{0, 1, ..., C - 1\}$.

The objective function of *standard adversarial training* (AT) (Madry et al., 2018) is

$$\min_{f_\theta \in \mathcal{F}} \frac{1}{n} \sum_{i=1}^n \ell(f_\theta(\tilde{x}_i), y_i), \tag{1}$$

where

$$\tilde{x}_i = \arg\max_{\tilde{x} \in \mathcal{B}_\epsilon[x_i]} \ell(f_\theta(\tilde{x}), y_i), \tag{2}$$

where $\tilde{x}$ is the most adversarial data within the $\epsilon$-ball centered at $x$, $f_\theta(\cdot) : \mathcal{X} \to \mathbb{R}^C$ is a score function, and the loss function $\ell : \mathbb{R}^C \times \mathcal{Y} \to \mathbb{R}$ is a composition of a base loss $\ell_{\text{B}} : \Delta^{C-1} \times \mathcal{Y} \to \mathbb{R}$ (e.g., the cross-entropy loss) and an inverse link function $\ell_{\text{L}} : \mathbb{R}^C \to \Delta^{C-1}$ (e.g., the soft-max activation), in which $\Delta^{C-1}$ is the corresponding probability simplex—in other words, $\ell(f_\theta(\cdot), y) = \ell_{\text{B}}(\ell_{\text{L}}(f_\theta(\cdot)), y)$. AT employs the *most adversarial data* generated according to Eq. (2) for updating the current model.

The objective function of *friendly adversarial training* (FAT) (Zhang et al., 2020b) is

$$\tilde{x}_i = \arg\min_{\tilde{x} \in \mathcal{B}_\epsilon[x_i]} \ell(f_\theta(\tilde{x}), y_i) \text{ s.t. } \ell(f_\theta(\tilde{x}), y_i) - \min_{y \in \mathcal{Y}} \ell(f_\theta(\tilde{x}), y) \geq \rho. \tag{3}$$

Note that the outer minimization remains the same as Eq. (1), and the operator $\arg\max$ is replaced by $\arg\min$. $\rho$ is a margin of loss values (i.e., the misclassification confidence). The constraint of Eq. (3) firstly ensures $\tilde{x}$ is misclassified, and secondly ensures for $\tilde{x}$ the wrong prediction is better than the desired prediction $y_i$ by at least $\rho$ in terms of the loss value. Among all such $\tilde{x}$ satisfying the constraint, Eq. (3) selects the one minimizing $\ell(f_\theta(\tilde{x}), y_i)$ by a violation of the value $\rho$. There are no constraints on $\tilde{x}_i$ if $\tilde{x}_i$ is correctly classified. FAT employs the *friendly adversarial data* generated according to Eq. (3) for updating the current model.

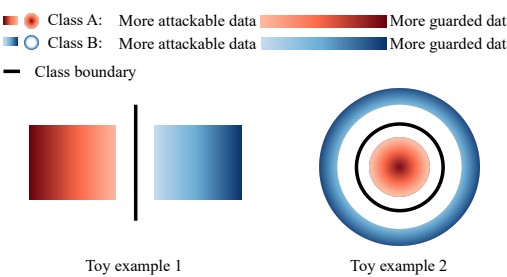 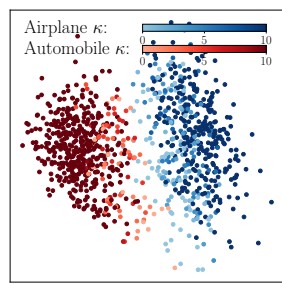

Figure 3: More attackable data (lighter red and blue) are closer to the class boundary; more guarded data (darker red and blue) are farther away from the class boundary. **Left panel**: Two toy examples. **Right panel**: The model's output distribution of two randomly selected classes from the CIFAR-10 dataset. The degree of robustness (denoted by the color gradient) of a data point is calculated based on the least number of iterations $\kappa$ that PGD needs to find its misclassified adversarial variant.

## 2.2 REALIZATIONS

AT and FAT's objective functions imply the optimization of adversarially robust networks, with one step generating adversarial data and one step minimizing loss on the generated adversarial data w.r.t. the model parameters $\theta$.

The projected gradient descent method (PGD) (Madry et al., 2018) is the most common approximation method for searching adversarial data. Given a starting point $x^{(0)} \in \mathcal{X}$ and step size $\alpha > 0$, PGD works as follows:

$$x^{(t+1)} = \Pi_{\mathcal{B}[x^{(0)}]}\big(x^{(t)} + \alpha \operatorname{sign}(\nabla_{x^{(t)}}\ell(f_\theta(x^{(t)}), y))\big), t \in \mathbb{N} \tag{4}$$

until a certain stopping criterion is satisfied. $\ell$ is the loss function; $x^{(0)}$ refers to natural data or natural data perturbed by a small Gaussian or uniformly random noise; $y$ is the corresponding label for natural data; $x^{(t)}$ is adversarial data at step $t$; and $\Pi_{\mathcal{B}_\epsilon[x_0]}(\cdot)$ is the projection function that projects the adversarial data back into the $\epsilon$-ball centered at $x^{(0)}$ if necessary.

There are different stopping criteria between AT and FAT. AT employs a fixed number of iterations $K$, namely, the PGD-$K$ algorithm (Madry et al., 2018), which is commonly used in many adversarial training methods such as CAT (Cai et al., 2018), DAT (Wang et al., 2019), TRADES (Zhang et al., 2019), and MART (Wang et al., 2020b). On the other hand, FAT employs the misclassification-aware criterion. For example, Zhang et al. (2020b) proposed the early-stopped PGD-$K$-$\tau$ algorithm ($\tau \leq K$; $K$ is the fixed and maximally allowed iteration number): Once the PGD-$K$-$\tau$ finds the current model misclassifying the adversarial data, it stops the iterations immediately ($\tau = 0$) or slides a few more steps ($\tau > 0$). This misclassification-aware criterion is used in the emerging adversarial training methods such as MMA (Ding et al., 2020), FAT (Zhang et al., 2020b), ATES (Sitawarin et al., 2020), and Customized AT (Cheng et al., 2020).

AT can enhance the robustness against adversarial data but, unfortunately, degrades the standard accuracy on the natural data significantly (Madry et al., 2018). On the other hand, FAT has better standard accuracy with near-zero or little degradation of robustness (Zhang et al., 2020b).

Nevertheless, both AT and FAT treat the generated adversarial data equally for updating the model parameters, which is not necessary and sometimes even pessimistic. In the next sections, we introduce our method GAIRAT, which is compatible with existing methods such as AT, FAT, and TRADES. Consequently, GAIRAT can significantly enhance robustness with little or even zero degradation of standard accuracy.

## 3 GEOMETRY-AWARE INSTANCE-REWEIGHTED ADVERSARIAL TRAINING

In this section, we propose geometry-aware instance-reweighted adversarial training (GAIRAT) and its learning objective as well as its algorithmic realization.

### 3.1 MOTIVATIONS OF GAIRAT

**Model capacity is often insufficient in adversarial training.** In the standard training, the over-parameterized networks, e.g., ResNet-18 and even larger ResNet-50, have more than enough model capacity, which can easily fit the natural training data entirely (Zhang et al., 2017). However, the left panel of Figure 2 shows that the model capacity of those over-parameterized networks is not enough for fitting the adversarial data. Under the computational budget of 100 epochs, the networks hardly reach zero error on the adversarial training data. Besides, adversarial training error only decreases by a small constant factor with the significant increase of the model's parameters. Even worse, a slightly larger perturbation bound $\epsilon_{\text{train}}$ significantly uncovers this insufficiency of the model capacity (right panel): Adversarial training error significantly increases with slightly larger $\epsilon_{\text{train}}$. Surprisingly, the standard training error on natural data hardly reaches zero with $\epsilon_{\text{train}} = 16/255$.

Adversarial training methods employ the adversarial data to reduce the sensitivity of the model's output w.r.t. small changes of the natural data (Papernot et al., 2016). During the training process, adversarial data are generated on the fly and are adaptively changed based on the current model to smooth the natural data's local neighborhoods. The volume of this surrounding is exponentially ($|1 + \epsilon_{\text{train}}|^{|\mathcal{X}|}$) large w.r.t. the input dimension $|\mathcal{X}|$, even if $\epsilon_{\text{train}}$ is small. Thus, this smoothness consumes significant model capacity. In adversarial training, we should carefully leverage the limited model capacity by fitting the important data and by ignoring the unimportant data.

**More attackable/guarded data are closer to/farther away from the class boundary.** We can measure the importance of the data by their robustness against adversarial attacks. Figure 3 shows that the robustness (more attackable or more guarded) of the data is closely related to their geometric distance from the decision boundary. From the geometry perspective, more attackable data are closer to the class boundary whose adversarial variants are more important to fine-tune the decision boundary for enhancing robustness.

Appendix A contains experimental details of Figures 2 and 3 and more motivation figures.

### 3.2 LEARNING OBJECTIVE OF GAIRAT

Let $\omega(x, y)$ be the *geometry-aware weight assignment function* on the loss of adversarial variant $\tilde{x}$. The inner optimization for generating $\tilde{x}$ still follows Eq. (2) or Eq. (3). The outer minimization is

$$\min_{f_\theta \in \mathcal{F}} \frac{1}{n} \sum_{i=1}^n \omega(x_i, y_i) \ell(f_\theta(\tilde{x}_i), y_i). \tag{5}$$

The constraint firstly ensures that $y_i = \arg\max_i f_\theta(x_i)$ and secondly ensures that $\omega(x_i, y_i)$ is a non-increasing function w.r.t. the *geometric distance*, i.e., the distance from data $x_i$ to the decision boundary, in which $\omega(x_i, y_i) \geq 0$ and $\frac{1}{n} \sum_{i=1}^n \omega(x_i, y_i) = 1$.

There are no constraints when $y_i \neq \arg\max_i f_\theta(x_i)$ : for those $x$ significantly far away from the decision boundary, we may discard them (outliers); for those $x$ close to the decision boundary, we may assign them large weights. In this paper, we do not consider outliers, and therefore we assign large weight to the losses of adversarial data, whose natural counterparts are misclassified. Figure 1 provides an illustrative schematic of the learning objective of GAIRAT.

A *burn-in period* may be introduced, i.e., during the initial period of the training epochs, $\omega(x_i, y_i) = 1$ regardless of the geometric distance of input $(x_i, y_i)$, because the geometric distance is less informative initially, when the classifier is not properly learned.

### 3.3 REALIZATION OF GAIRAT

The learning objective Eq. (5) implies the optimization of an adversarially robust network, with one step generating adversarial data and then reweighting loss on them according to the geometric distance of their natural counterparts, and one step minimizing the reweighted loss w.r.t. the model parameters $\theta$.

We approximate the geometric distance of a data point $(x, y)$ by the least iteration numbers $\kappa(x, y)$ that the PGD method needs to generate a adversarial variant $\tilde{x}$ to fool the current network, given the

---

**Algorithm 1** Geometry-aware projected gradient descent (GA-PGD)

---

**Input:** data $x \in \mathcal{X}$, label $y \in \mathcal{Y}$, model $f$, loss function $\ell$, maximum PGD step $K$, perturbation bound $\epsilon$, step size $\alpha$
**Output:** adversarial data $\tilde{x}$ and geometry value $\kappa(x, y)$
$\tilde{x} \leftarrow x; \kappa(x, y) \leftarrow 0$
**while** $K > 0$ **do**
   **if** $\arg\max_i f(\tilde{x}) = y$ **then**
      $\kappa(x, y) \leftarrow \kappa(x, y) + 1$
   **end if**
   $\tilde{x} \leftarrow \Pi_{\mathcal{B}[x,\epsilon]}\big(\alpha \operatorname{sign}(\nabla_{\tilde{x}}\ell(f(\tilde{x}), y)) + \tilde{x}\big)$
   $K \leftarrow K - 1$
**end while**

---

**Algorithm 2** Geometry-aware instance-dependent adversarial training (GAIRAT)

---

**Input:** network $f_\theta$, training dataset $S = \{(x_i, y_i)\}_{i=1}^n$, learning rate $\eta$, number of epochs $T$, batch size $m$, number of batches $M$
**Output:** adversarially robust network $f_\theta$
**for** epoch $= 1, \ldots, T$ **do**
   **for** mini-batch $= 1, \ldots, M$ **do**
      Sample a mini-batch $\{(x_i, y_i)\}_{i=1}^m$ from $S$
      **for** $i = 1, \ldots, m$ (in parallel) **do**
         Obtain adversarial data $\tilde{x}_i$ of $x_i$ and geometry value $\kappa(x_i, y_i)$ by Algorithm 1
         Calculate $\omega(x_i, y_i)$ according to geometry value $\kappa(x_i, y_i)$ by Eq. 6
      **end for**
      $\theta \leftarrow \theta - \eta \nabla_\theta \left\{ \sum_{i=1}^m \frac{\omega(x_i, y_i)}{\sum_{j=1}^m \omega(x_j, y_j)} \ell(f_\theta(\tilde{x}_i), y_i) \right\}$
   **end for**
**end for**

---

maximally allowed iteration number $K$ and step size $\alpha$. Thus, the geometric distance is approximated by $\kappa$ (precisely by $\kappa \times \alpha$). Thus, the value of the weight function $\omega$ should be non-increasing w.r.t. $\kappa$. We name $\kappa(x, y)$ the *geometry value* of data $(x, y)$.

How to calculate the optimal $\omega$ is still an open question; therefore, we heuristically design different non-increasing functions $\omega$. We give one example here and discuss more examples in Appendix C.3 and Section 4.1.

$$w(x, y) = \frac{(1 + \tanh(\lambda + 5 \times (1 - 2 \times \kappa(x, y)/K)))}{2}, \tag{6}$$

where $\kappa/K \in [0, 1]$, $K \in \mathbb{N}^+$, and $\lambda \in \mathbb{R}$. If $\lambda = +\infty$, GAIRAT recovers the standard AT (Madry et al., 2018), assigning equal weights to the losses of adversarial data.

Algorithm 1 is a geometry-aware PGD method (GA-PGD), which returns both the most adversarial data and the geometry value of its natural counterpart. Algorithm 2 is geometry-aware instance-dependent adversarial training (GAIRAT). GAIRAT leverages Algorithms 1 for obtaining the adversarial data and the geometry value. For each mini-batch, GAIRAT reweighs the loss of adversarial data $(\tilde{x}_i, y_i)$ according to the geometry value of their natural counterparts $(x_i, y_i)$, and then updates the model parameters by minimizing the sum of the reweighted loss.

GAIRAT is a general method. Indeed, FAT (Zhang et al., 2020b) and TRADES (Zhang et al., 2019) can be modified to GAIR-FAT and GAIR-TRADES (see Appendices B.1 and B.2, respectively).

**Comparisons with SVM.** The abstract concept of GAIRAT has appeared previously. For example, in the support vector machine (SVM), support vectors near the decision boundary are particularly useful in influencing the decision boundary (Hearst et al., 1998). For learning models, the magnitude of the loss function (e.g., the hinge loss and the logistic loss) can naturally capture different data's geometric distance from the decision boundary. For updating the model, the loss function treats data differently by incurring large losses on important attackable (close to the decision bound-

ary) or misclassified data and incurring zero or very small losses on unimportant guarded (far away from the decision boundary) data.

However, in adversarial training, it is critical to explicitly assign different weights on top of losses on different adversarial data due to the *blocking effect*: The model trained on the adversarial data that maximize the loss learns to prevent generating large-loss adversarial data. This blocking effect makes the magnitude of the loss less capable of distinguishing important adversarial data from unimportant ones for updating the model parameters, compared with the role of loss on measuring the natural data's importance in standard training. Our GAIRAT breaks this blocking effect by explicitly extracting data's geometric information to distinguish the different importance.

**Comparisons with AdaBoost and focal loss.** The idea of instance-dependent weighting has been studied in the literature. Besides robust estimator (e.g., M-estimator (Boos & Stefanski, 2013)) for learning under outliers (e.g., label-noised data), hard data mining is another branch where our GAIRAT belongs. Boosting algorithms such as AdaBoost (Freund & Schapire, 1997) select harder examples to train subsequent classifiers. Focal loss (Lin et al., 2017) is specially designed loss function for mining hard data and misclassified data. However, the previous hard data mining methods leverage the data's losses for measuring the hardness; by comparison, our GAIRAT measures the hardness by how difficulty the natural data are attacked (i.e., geometry value $\kappa$). This new measurement $\kappa$ sheds new lights on measuring the data's hardness (Zhu et al., 2021).

**Comparisons with related adversarial training methods.** Some existing adversarial training methods also "treat adversarial data differently", but in different ways to our GAIRAT. For example, CAT (Cai et al., 2018), MMA (Ding et al., 2020), and DAT (Wang et al., 2019) methods generate the differently adversarial data for updating model over the training process. CAT utilized the adversarial data with different PGD iterations $K$. DAT utilized the adversarial data with different convergence qualities. MMA leveraged adversarial data with instance-dependent perturbation bounds $\epsilon$. Different from those existing methods, our GAIRAT treat adversarial data differently by explicitly assigning different weights on their losses, which can break the blocking effect.

Note that the learning objective of MART (Wang et al., 2020b) also explicitly assigns weights, not directly on the adversarial loss but KL divergence loss (see details in Section C.7). The KL divergence loss helps to strengthen the smoothness within the norm ball of natural data, which is also used in VAT (Miyato et al., 2016) and TRADES (Zhang et al., 2019). Differently from MART, our GAIRAT explicitly assigns weights on the adversarial loss. Therefore, we can easily modify MART to GAIR-MART (see experimental comparisons in Section C.7). Besides, MART assigns weights based on the model's prediction confidence on the natural data; GAIRAT assigns weights based on how easy the natural data can be attacked (geometry value $\kappa$).

**Comparisons with the geometric studies of DNN.** Researchers in adversarial robustness employed the first-order or second-order derivatives w.r.t. input data to explore the DNN's geometric properties (Fawzi et al., 2017; Kanbak et al., 2018; Fawzi et al., 2018; Qin et al., 2019; Moosavi-Dezfooli et al., 2019). Instead, we have a complementary but different argument: Data points themselves are geometrically different regardless of DNN. The geometry value $\kappa$ in adversarial training (AT) is an approximated measurement of data's geometric properties due to the AT's smoothing effect (Zhu et al., 2021).

## 4 EXPERIMENTS

In this section, we empirically justify the efficacy of GAIRAT. Section 4.1 shows that GAIRAT can relieve the undesirable robust overfitting (Rice et al., 2020) of the minimax-based adversarial training (Madry et al., 2018). Note that some concurrent studies (Chen et al., 2021a;b) provided various adversarial training strategies, which can also mitigate the issue of robust overfitting. In Section 4.2, we benchmark our GAIRAT and GAIR-FAT using Wide ResNets and compare them with AT and FAT.

In our experiments, we consider $||\tilde{x} - x||_\infty \leq \epsilon$ with the same $\epsilon$ in both training and evaluations. All images of CIFAR-10 (Krizhevsky, 2009) and SVHN (Netzer et al., 2011) are normalized into $[0, 1]$.

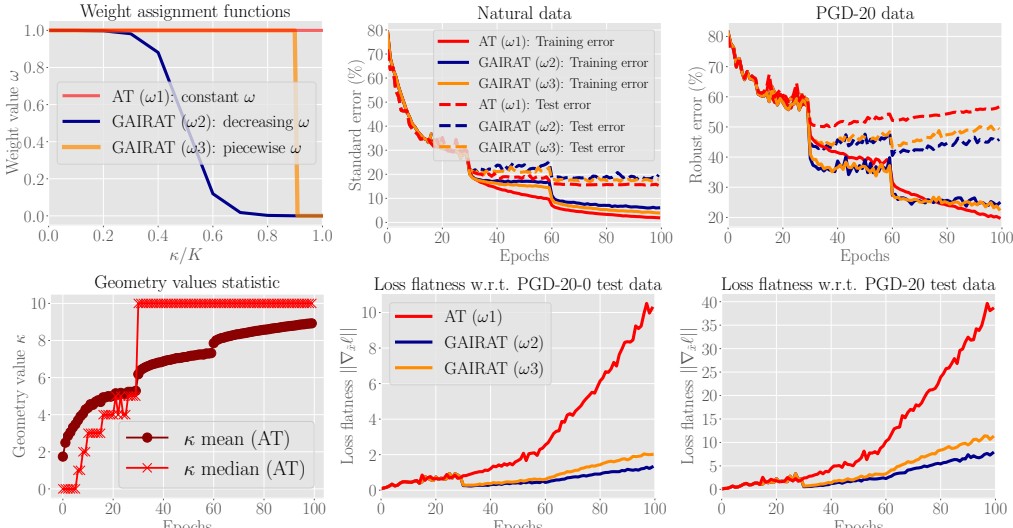

Figure 4: Comparisons of AT ($\omega_1$, red lines) and GAIRAT ($\omega_2$, blue lines and $\omega_3$, yellow lines) using ResNet-18 on the CIFAR-10 dataset. **Upper-left panel** shows different weight assignment functions $\omega$ w.r.t. the geometry value $\kappa$. **Bottom-left panel** reports the training statistic of the standard AT and calculates the median (dark red circle) and mean (light red cross) of geometry values of all training data at each epoch. **Upper-middle and upper-right panels** report standard training/test errors and robust training/test errors, respectively. **Bottom-middle and bottom-right panels** report the loss flatness w.r.t. friendly adversarial test data and most adversarial test data, respectively.

## 4.1 GAIRAT RELIEVES ROBUST OVERFITTING

In Figure 4, we conduct the standard AT (all red lines) using ResNet-18 (He et al., 2016) on CIFAR-10 dataset. For generating the most adversarial data for updating the model, the perturbation bound $\epsilon = 8/255$; the PGD steps number $K = 10$ with step size $\alpha = 2/255$, which keeps the same as Rice et al. (2020). We train ResNet-18 using SGD with 0.9 momentum for 100 epochs with the initial learning rate of 0.1 divided by 10 at Epoch 30 and 60, respectively. At each training epoch, we collect the training statistics, i.e., the geometry value $\kappa(x, y)$ of each training data, standard/robust training and test error, the flatness of loss w.r.t. adversarial test data. The detailed descriptions of those statistics and the evaluations are in the Appendix C.1.

Bottom-left panel of Figure 4 shows geometry value $\kappa$ of training data of standard AT. Over the training progression, there is an increasing number of guarded training data with a sudden leap when the learning rate decays to 0.01 at Epoch 30. After Epoch 30, the model steadily engenders a increasing number of guarded data whose adversarial variants are correctly classified. Learning from those correctly classified adversarial data (large portion) will reinforce the existing knowledge and spare little focus on wrongly predicted adversarial data (small portion), thus leading to the robust overfitting. The robust overfitting is manifested by red (dashed and solid) lines in upper-middle and upper-right and bottom-middle and bottom-right panels.

To avoid the large portion of guarded data overwhelming the learning from the rare attackable data, our GAIRAT explicitly give small weights to the losses of adversarial variants of the guarded data. Blue ($\omega_2$) and yellow ($\omega_3$) lines in upper-left panel give two types of weight assignment functions that assign instance-dependent weight on the loss based on the geometry value $\kappa$. In GAIRAT, the model is forced to give enough focus on those rare attackable data.

In GAIRAT, the initial 30 epochs is burn-in period, and we introduce the instance-dependent weight assignment $\omega$ from Epoch 31 onward (both blue and yellow lines in Figure 4). The rest of hyperparameters keeps the same as AT (red lines). From the upper-right panel, GAIRAT (both yellow and blue lines) achieves smaller error on adversarial test data and larger error on training adversarial data, compared with standard AT (red lines). Therefore, our GAIRAT can relieve the issue of the robust overfitting.

Besides, Appendix C contains more experiments such as different learning rate schedules, different choices of weight assignment functions $\omega$, different lengths of burn-in period, a different dataset (SVHN) and different networks (Small CNN and VGG), which all justify the efficacy of our GAIRAT. Notably, in Appendix C.6, we show the effects of GAIR-FAT on improving FAT.

## 4.2 PERFORMANCE EVALUATION ON WIDE RESNETS

Table 1: Test accuracy of WRN-32-10 on CIFAR-10 dataset

| Defense | Best checkpoint | | | | | | Last checkpoint | | | | | |
|---|---|---|---|---|---|---|---|---|---|---|---|---|
| | Natural | Diff. | PGD-20 | Diff. | PGD+ | Diff. | Natural | Diff. | PGD-20 | Diff. | PGD+ | Diff. |
| AT | $86.92 \pm 0.24$ | - | $51.96 \pm 0.21$ | - | $51.28 \pm 0.23$ | - | $86.62 \pm 0.22$ | - | $46.73 \pm 0.08$ | - | $46.08 \pm 0.07$ | - |
| FAT | $\mathbf{89.16 \pm 0.15}$ | $+2.24$ | $51.24 \pm 0.14$ | $-0.72$ | $46.14 \pm 0.19$ | $-5.14$ | $88.18 \pm 0.19$ | $+1.56$ | $46.79 \pm 0.34$ | $+0.06$ | $45.80 \pm 0.16$ | $-0.28$ |
| GAIRAT | $85.75 \pm 0.23$ | $-1.17$ | $\mathbf{57.81 \pm 0.54}$ | $+5.85$ | $\mathbf{55.61 \pm 0.61}$ | $+4.33$ | $85.49 \pm 0.25$ | $-1.13$ | $\mathbf{53.76 \pm 0.49}$ | $+7.03$ | $\mathbf{50.32 \pm 0.48}$ | $+4.24$ |
| GAIR-FAT | $88.59 \pm 0.12$ | $+1.67$ | $56.21 \pm 0.52$ | $+4.25$ | $53.50 \pm 0.60$ | $+2.22$ | $\mathbf{88.44 \pm 0.10}$ | $+1.82$ | $50.64 \pm 0.56$ | $+3.91$ | $47.51 \pm 0.51$ | $+1.43$ |

We employ the large-capacity network, i.e., Wide ResNet (Zagoruyko & Komodakis, 2016), on the CIFAR-10 dataset. In Table 1, we compare the performance of the standard AT (Madry et al., 2018), FAT (Zhang et al., 2020b), GAIRAT and GAIR-FAT. We use WRN-32-10 that keeps the same as Madry et al. (2018). We compare different methods on the best checkpoint model (suggested by Rice et al. (2020)) and the last checkpoint model (used by Madry et al. (2018)), respectively. Note that results in Zhang et al. (2020b) only compare the last checkpoint between AT and FAT; instead, we also include the best checkpoint comparisons. We evaluate the robust models based on the three evaluation metrics, i.e., standard test accuracy on natural data (Natural), robust test accuracy on adversarial data generated by PGD-20 and PGD+. PGD+ is PGD with five random starts, and each start has 40 steps with step size 0.01, which keeps the same as Carmon et al. (2019) (PGD+ has $40 \times 5 = 200$ iterations for each test data). We run AT, FAT, GAIRAT, and GAIR-FAT five repeated trials with different random seeds. Table 1 reports the medians and standard deviations of the results. Besides, we treat the results of AT as the baseline and report the difference (Diff.) of the test accuracies. The detailed training settings and evaluations are in Appendix C.8. Besides, we also compare TRADES and GAIR-TRADES using WRN-34-10, which is in the Appendix C.9.

Compared with standard AT, our GAIRAT significantly boosts adversarial robustness with little degradation of accuracy, which challenges the inherent trade-off. Besides, FAT also challenges the inherent trade-off instead by improving accuracy with little degradation of robustness. Combining two directions, i.e., GAIR-FAT, we can improve both robustness and accuracy of standard AT. Therefore, Table 1 affirmatively confirms the efficacy of our geometry-aware instance-reweighted methods in significantly improving adversarial training.

## 5 CONCLUSION AND FUTURE WORK

This paper has proposed a novel adversarial training method, i.e., geometry-aware instance-reweighted adversarial training (GAIRAT). GAIRAT gives more (less) weights to loss of the adversarial data whose natural counterparts are closer to (farther away from) the decision boundary. Under the limited model capacity and the inherent inequality of the data, GAIRAT sheds new lights on improving the adversarial training.

GAIRAT training under the PGD attacks can defend PGD attacks very well, but indeed, it cannot perform equally well on all existing attacks (Chen et al., 2021a). From the philosophical perspective, we cannot expect defenses under one specific attack can defend all existing attacks, which echoes the previous finding that "it is essential to include adversarial data produced by all known attacks, as the defensive training is non-adaptive (Papernot et al., 2016)." Incorporating all attacks in GAIRAT yet preserving the efficiency is an interesting future direction. Besides, it still an open question to design the optimal weight assignment function $\omega$ in Eq. 5 or to design a proper network structure suitable to adversarial training. Furthermore, there is still a large room to apply adversarial training techniques into other domains such as pre-training (Hendrycks et al., 2019; Chen et al., 2020; Jiang et al., 2020; Salman et al., 2020), noisy labels (Zhu et al., 2021) and so on.

ACKNOWLEDGMENT

JZ, GN, and MS were supported by JST AIP Acceleration Research Grant Number JPMJCR20U3, Japan. MS was also supported by the Institute for AI and Beyond, UTokyo. JNZ and BH were supported by the HKBU CSD Departmental Incentive Scheme. BH was supported by the RGC Early Career Scheme No. 22200720 and NSFC Young Scientists Fund No. 62006202. MK was supported by the National Research Foundation, Singapore under its Strategic Capability Research Centres Funding Initiative.

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

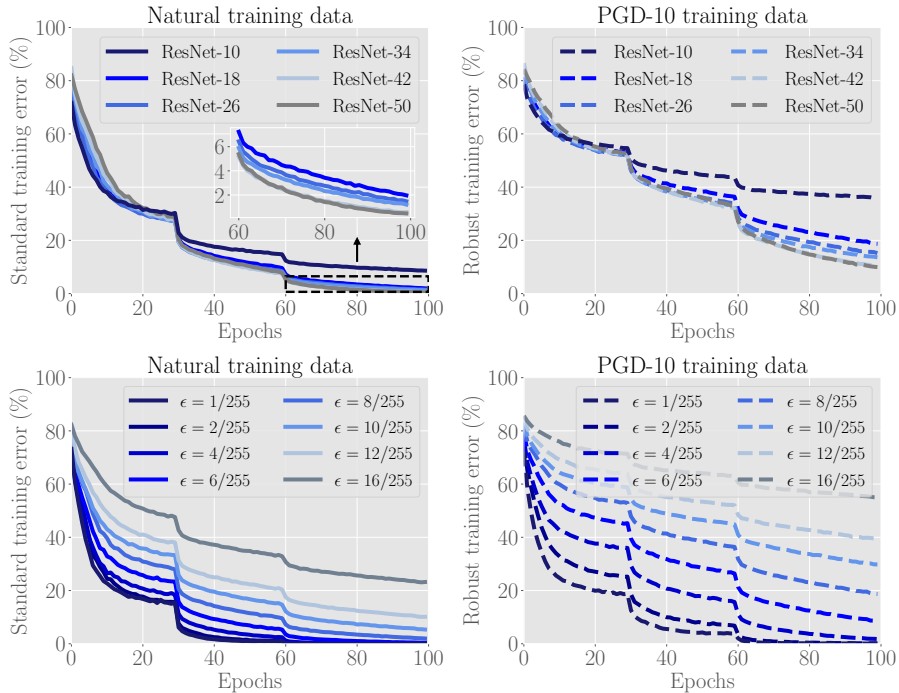

Figure 5: We plot standard training error (the left two panels) and adversarial training error (the right two panels) over the training epochs of the standard AT on CIFAR-10 dataset. Top two panels: standard AT on different sizes of network. Bottom two panels: standard AT on ResNet-18 under different perturbation bound $\epsilon_{\mathrm{train}}$.

## A  MOTIVATIONS OF GAIRAT

We show that model capacity is often insufficient in adversarial training, especially when $\epsilon_{\mathrm{train}}$ is large; therefore, the model capacity should be carefully preserved for fitting important data.

In this section, we give experimental details of Figure 2 and provide complementary experiments in Figures 5 and 6. In the left panel of Figure 2 and top two panels of Figure 5, we use standard AT to train different sizes of network under the perturbation bound $\epsilon_{\mathrm{train}} = 8/255$ on CIFAR-10 dataset. In the right panel of Figure 2 and two bottom panels of Figure 5, we fix the size of network and use ResNet-18; we conduct standard AT under different values of perturbation bound $\epsilon_{\mathrm{train}} \in [1/255, 16/255]$. The solid lines show the standard training error on natural data and the dash lines show the robust training error on adversarial training data.

**Training details**  We train all the different networks for 100 epochs using SGD with 0.9 momentum. The initial learning rate is 0.1, reduced to 0.01, 0.001 at Epoch 30, and 60, respectively. The weight decay is 0.0005. For generating the most adversarial data for updating the model, we use the PGD-10 attack. The PGD steps number $K = 10$ and the step size $\alpha = \epsilon/4$. There is a random start, i.e., uniformly random perturbations ($[-\epsilon_{\mathrm{train}}, +\epsilon_{\mathrm{train}}]$) added to natural data before PGD perturbations for generating PGD-10 training data. We report the standard training error on the natural training data and the robust training error on the adversarial training data that are generated by the PGD-10 attack.

We also conduct the experiments on the SVHN dataset in Figure 6. The training setting keeps the same as that of CIFAR-10 experiments except using 0.01 as the initial learning rate, reduced to 0.001, 0.0001 at Epoch 30, and 60, respectively. We find standard AT always fails when the perturbation bound is larger than $\epsilon = 16/255$ for the SVHN dataset due to the severe cross-over mixture issue (Zhang et al., 2020b); therefore, we do not report its results.

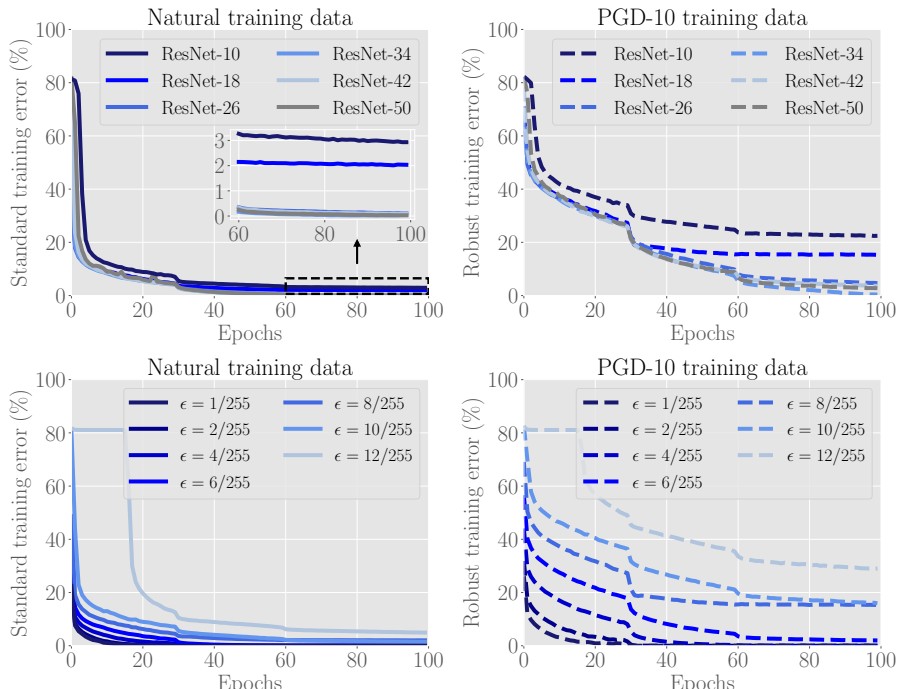

Figure 6: We plot standard training error (the left two panels) and adversarial training error (the right two panels) over the training epochs of the standard AT on **SVHN** dataset. Top two panels: AT on different sizes of network. Bottom two panels: AT on ResNet-18 under different perturbation bound $\epsilon_{\text{train}}$.

Next, we show that more attackable (more important) data are closer to the decision boundary; more guarded (less important) data are farther away from the decision boundary.

In Figures 7 and 8, we plot 2-d visualizations of the output distributions of a robust ResNet-18 on CIFAR-10 dataset. We take the robust ResNet-18 at the checkpoint of Epoch 30 (red line in Figure 9) as our base model here. For each class in the CIFAR-10 dataset, we randomly sample 1000 training datapoints for visualization. For each data point, we compute its the least number of iterations $\kappa$ that PGD requires to find its misclassified adversarial variant. For PGD, we set the perturbation bound $\epsilon = 0.031$, the step size $\alpha = 0.31/4$, and the maximum PGD steps $K = 10$. Then, each data point has its unique robustness attribution, i.e., value $\kappa$. We take those data as the input of the robust ResNet and output 10-dimensional logits, and then, we use principal components analysis (PCA) to project 10-dimensional logits into 2-dimension for visualization. The color gradient denotes the degree of the robustness of each data point. The more attackable data have lighter colors (red or blue), and the more guarded data has darker colors (red or blue).

From Figures 7 and 8, we find that the attackable data in general are geometrically close to the decision boundary while the guarded data in general are geometrically far away from the decision boundary. It is also very interesting to observe that not all classes are well separated. For example, Cat-Dog is less separable than Cat-Ship in second row of Figure 8.

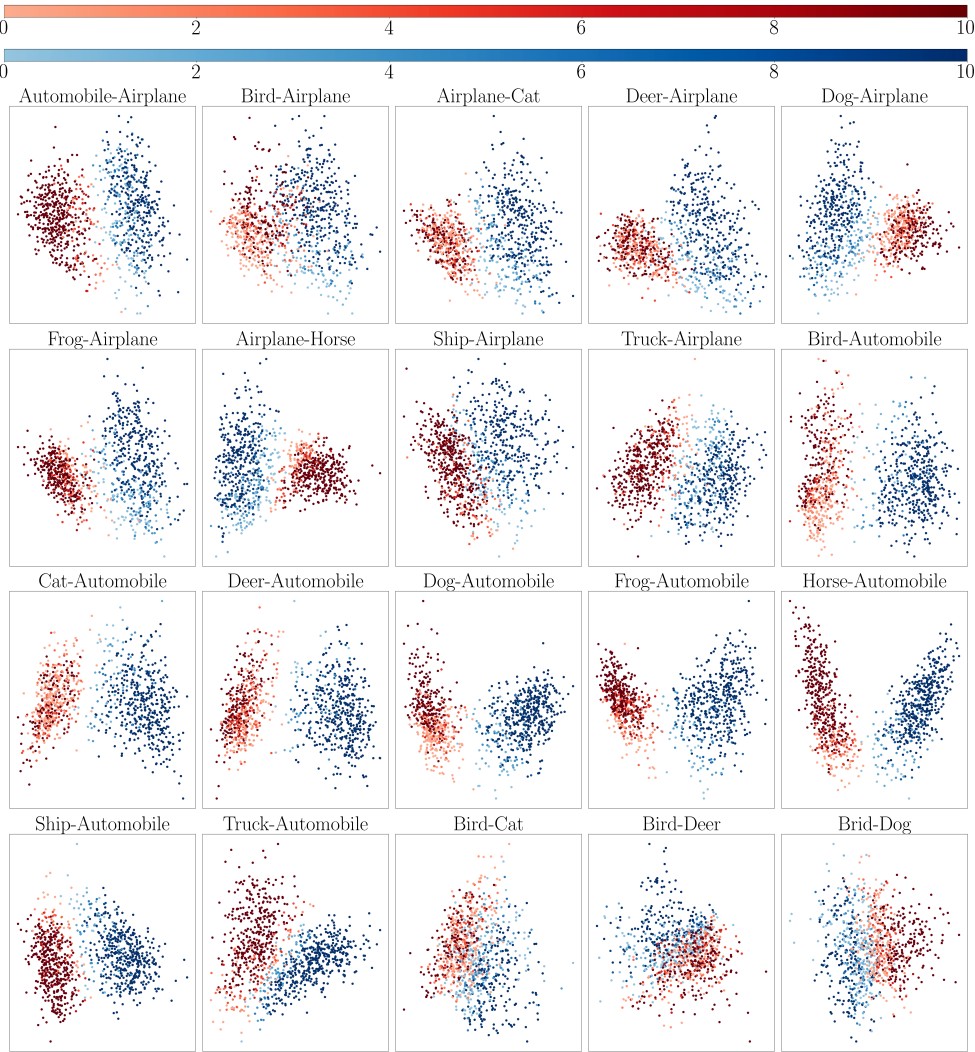

Figure 7: **Part A** 2-d visualizations of the model's output distribution of natural training data from two separated classes from CIFAR-10 dataset. The degree of the robustness (denoted by the color gradient) of a datum is calculated based on the least number of iterations $\kappa$ that PGD requires to find its misclassified adversarial variant. The light blue and light red points represent attackable data which are close to the class boundary; the dark blue and dark red points represent the guarded data which are far away from the decision boundary. (Top colorbars corresponds to the value $\kappa$)

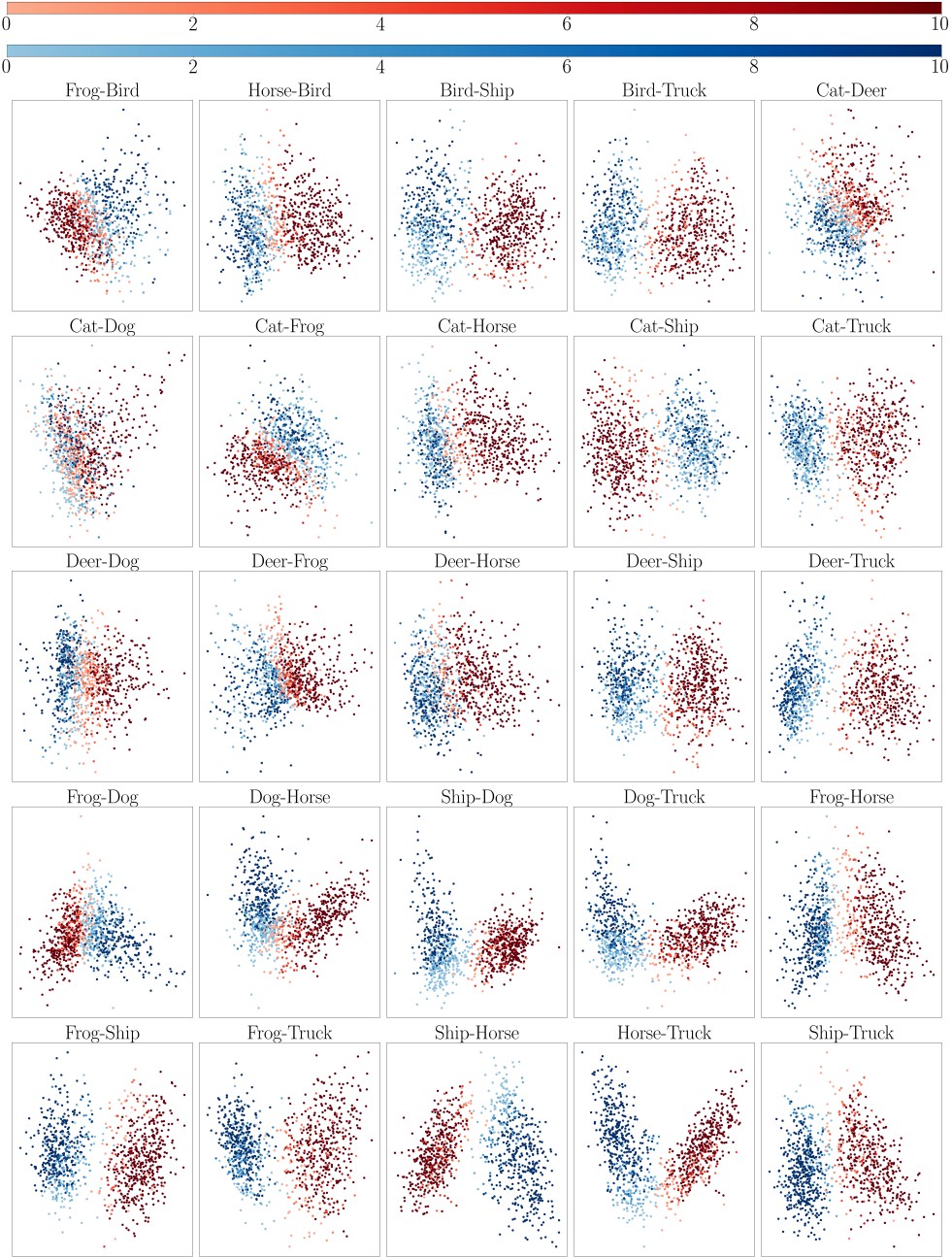

Figure 8: **Part B** - 2-d visualizations of the model's output distribution on CIFAR-10 dataset.

# B ALGORITHMS

## B.1 GEOMETRY-AWARE INSTANCE-REWEIGHTED FRIENDLY ADVERSARIAL TRAINING (GAIR-FAT)

---

**Algorithm 3** Geometry-aware early stopped PGD-$K$-$\tau$

---

**Input:** data $x \in \mathcal{X}$, label $y \in \mathcal{Y}$, model $f$, loss function $\ell$, maximum PGD step $K$, step $\tau$, perturbation bound $\epsilon$, step size $\alpha$
**Output:** friendly adversarial data $\tilde{x}$ and geometry value $\kappa(x, y)$
$\tilde{x} \leftarrow x; \kappa(x, y) \leftarrow 0$
**while** $K > 0$ **do**
  **if** $\arg\max_i f(\tilde{x}) \neq y$ and $\tau = 0$ **then**
    **break**
  **else if** $\arg\max_i f(\tilde{x}) \neq y$ **then**
    $\tau \leftarrow \tau - 1$
  **else**
    $\kappa(x, y) \leftarrow \kappa(x, y) + 1$
  **end if**
  $\tilde{x} \leftarrow \Pi_{\mathcal{B}[x,\epsilon]}\big(\alpha \operatorname{sign}(\nabla_{\tilde{x}} \ell(f(\tilde{x}), y)) + \tilde{x}\big)$
  $K \leftarrow K - 1$
**end while**

---

GAIRAT is a general method, and the friendly adversarial training (Zhang et al., 2020b) can be easily modified to a geometry-aware instance-reweighted version, i.e. GAIR-FAT.

GAIR-FAT utilizes Algorithm 3 to generate friendly adversarial data $(\tilde{x}, y)$ and the corresponding geometry value $\kappa(x, y)$, and then utilizes Algorithm 2 to update the model parameters.

### B.2 GEOMETRY-AWARE INSTANCE-REWEIGHTED TRADES (GAIR-TRADES)

---

**Algorithm 4** Geometry-aware PGD for TRADES

---

**Input:** data $x \in \mathcal{X}$, label $y \in \mathcal{Y}$, model $f$, loss function $\ell_{KL}$, maximum PGD step $K$, perturbation bound $\epsilon$, step size $\alpha$
**Output:** adversarial data $\tilde{x}$ and geometry value $\kappa(x, y)$
$\tilde{x} \leftarrow x + \xi \mathcal{N}(\mathbf{0}, \mathbf{I}); \kappa(x, y) \leftarrow 0$
**while** $K > 0$ **do**
    **if** $\arg \max_i f(\tilde{x}) = y$ **then**
        $\kappa(x, y) \leftarrow \kappa(x, y) + 1$
    **end if**
    $\tilde{x} \leftarrow \Pi_{\mathcal{B}[x, \epsilon]} \big( \alpha \operatorname{sign}(\nabla_{\tilde{x}} \ell_{KL}(f(\tilde{x}), f(x)) + \tilde{x} \big)$
    $K \leftarrow K - 1$
**end while**

---

**Algorithm 5** Geometry-aware instance-reweighted TRADES (GAIR-TRADES)

---

**Input:** network $f_\theta$, training dataset $S = \{(x_i, y_i)\}_{i=1}^n$, learning rate $\eta$, number of epochs $T$, batch size $m$, number of batches $M$
**Output:** adversarially robust network $f_\theta$
**for** epoch $= 1, \ldots, T$ **do**
    **for** mini-batch $= 1, \ldots, M$ **do**
        Sample a mini-batch $\{(x_i, y_i)\}_{i=1}^m$ from $S$
        **for** $i = 1, \ldots, m$ (in parallel) **do**
            Obtain adversarial data $\tilde{x}_i$ of $x_i$ and geometry value $\kappa(x_i, y_i)$ by Algorithm 4
            Calculate $\omega(x_i, y_i)$ according to geometry value $\kappa(x_i, y_i)$ by Eq. (6)
        **end for**
        Calculate the normalized $\omega_i = \frac{\omega(x_i, y_i)}{\sum_{j=1}^m \omega(x_j, y_j)}$ for each data

$$\theta \leftarrow \theta - \eta \nabla_\theta \sum_{i=1}^m \left\{ \omega_i \ell_{CE}(f_\theta(x_i), y_i) + \beta \ell_{KL}(f_\theta(\tilde{x}_i), f_\theta(x_i)) \right\}$$

    **end for**
**end for**

---

We modify TRADES (Zhang et al., 2019) to a GAIRAT version, i.e. GAIR-TRADES (Algorithms 4 and 5). Different from GAIRAT and GAIR-FAT, GAIR-TRADES employs Algorithm 4 to generate adversarial data $(\tilde{x}, y)$ and the corresponding geometry value $\kappa(x, y)$, and then utilizes both natural data and their adversarial variants to update the model parameters (Algorithm 5). Note that TRADES utilizes *virtual adversarial data* (Miyato et al., 2016) for updating the current model. The generated virtual adversarial data do not require any label information; therefore, their supervision signals heavily rely on their natural counterparts. Thus, in GAIR-TRADES, the instance-reweighting function $\omega$ applies to the loss of their natural data.

In Algorithm 4, $\mathcal{N}(\mathbf{0}, \mathbf{I})$ generates a random unit vector. $\xi$ is a small constant. $\ell_{KL}$ is Kullback-Leibler loss. In Algorithm 5, $\beta > 0$ is a regularization parameter for TRADES. $\ell_{CE}$ is cross-entropy loss. $\ell_{KL}$ is Kullback-Leibler loss, which keeps the same as Zhang et al. (2019).

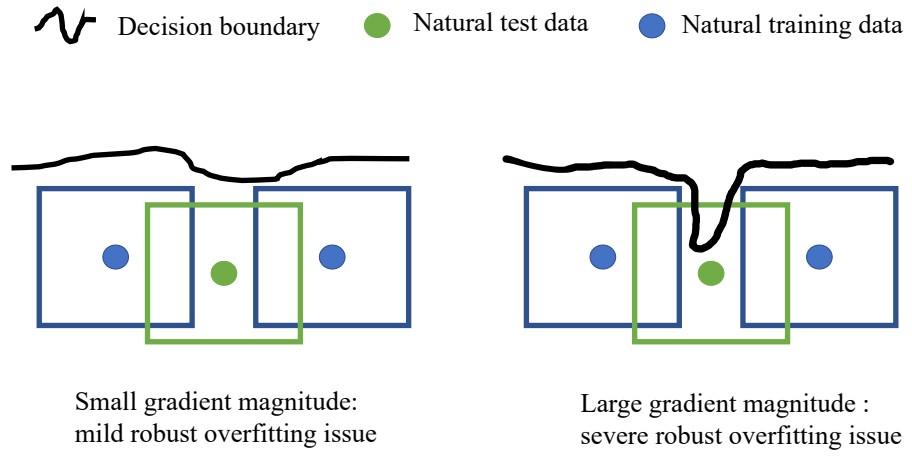

Figure 9: Illustration of the reasons for the issue of robust overfitting.

## C    EXTENSIVE EXPERIMENTS

### C.1    GAIRAT RELIEVES ROBUST OVERFITTING

In this section, we give the detailed descriptions of Figure 4 and provide more analysis and complementary experiments using the SVHN dataset in Figure 10.

In Figure 4, red lines (solid and dashed lines) refer to standard adversarial training (AT) (Madry et al., 2018). Blue and yellow lines (solid and dashed) refer to our geometry-aware instance-reweighted adversarial training (GAIRAT). Blue lines represent that GAIRAT utilizes the decreasing $\omega$ for assigning instance-dependent weights (corresponding to the blue line in the bottom-left panel); yellow lines represent that GAIRAT utilizes the non-increasing piece-wise $\omega$ for assigning instance-dependent weights (corresponding to the yellow line in the bottom-left panel). In the upper-left panel of Figure 4, we calculate the mean and median of geometry values $\kappa(x, y)$ of all 50K training data at each epoch. Geometry value $\kappa(x, y)$ of data $(x, y)$ refers to the least number of PGD steps that PGD methods need to generate a misclassified adversarial variant. Note that when the natural data is misclassified without any adversarial perturbations, the geometry value $\kappa(x, y) = 0$. The bottom-left panel calculates the instance-dependent weight $\omega$ for the loss of adversarial data based on the geometry value $\kappa$.

In the upper-middle panel of Figure 4, the solid lines represent the standard training error on the natural training data; the dashed lines represent the standard test error on the natural test data.

In the upper-right panel of Figure 4, the solid lines represent the robust training error on the adversarial training data; the dashed lines represent the robust test error on the adversarial test data. The adversarial training/test data are generated by PGD-20 attack with random start. Random start refers to the uniformly random perturbation of $[-\epsilon, \epsilon]$ added to the natural data before PGD perturbations. The step size $\alpha = 2/255$, which is the same as Wang et al. (2019).

In the bottom-middle and bottom-right panels of Figure 4, we calculate the flatness of the adversarial loss $\ell(f_\theta(\tilde{x}), \tilde{x})$ w.r.t. the adversarial data $\tilde{x}$. In the bottom-middle panel, adversarial data refer to the friendly adversarial test data that are generated by early-stopped PGD-20-0 (Zhang et al., 2020b). The maximum PGD step number is 20; $\tau = 0$ means the immediate stop once the wrongly predicted adversarial test data are found. We use friendly adversarial test data to approximate the points on decision boundary of the robust model $f_\theta$. The flatness of the decision boundary is approximated by average of $||\nabla_{\tilde{x}}\ell||$ across all 10K adversarial test data. We give the flatness value at each training epoch (higher flatness value refers to higher curved decision boundary, see Figure 9).

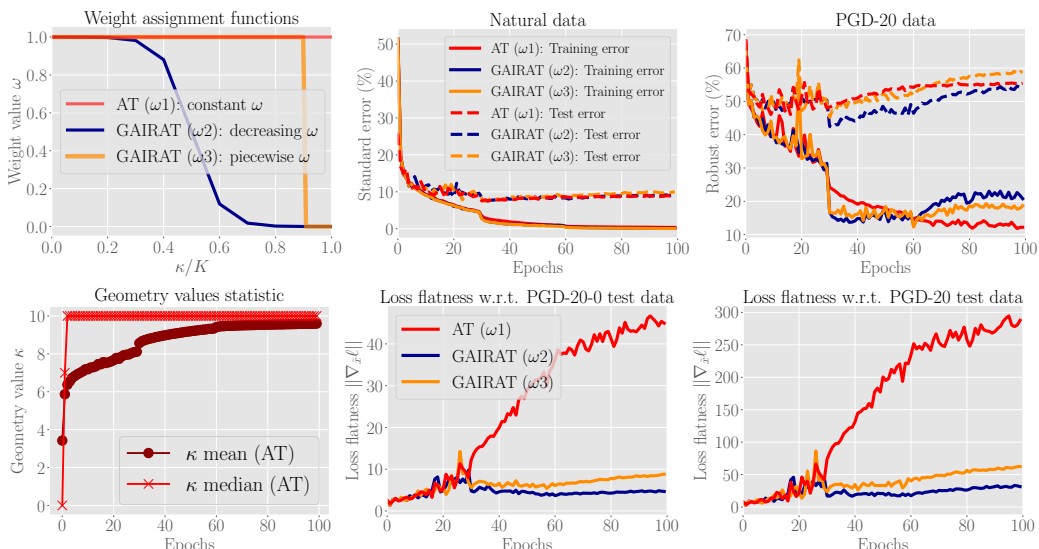

Figure 10: Comparisons of AT ($\omega_1$, red lines) and GAIRAT ($\omega_2$, blue lines and $\omega_3$, yellow lines) using ResNet-18 on **SVHN** dataset. **Upper-left panel** shows different instance-dependent weight assignment functions $\omega$ w.r.t. the geometry value $\kappa$. **Bottom-left panel** reports the standard AT training statistic and calculates the median (dark red circle) and mean (light red cross) of geometry values of all training data at each epoch. **Upper-middle and upper-right panels** report natural training/test errors and robust training/test errors, respectively. **Bottom-middle and bottom-right panels** report the loss flatness w.r.t. friendly adversarial test data and most adversarial test data, respectively.

For completeness, the bottom-right panel uses the most adversarial test data that are generated by PGD-20 (Madry et al., 2018).

The magnitude of the norm of gradients, i.e., $||\nabla_{\tilde{x}}\ell||$, is a reasonable metric for measuring the magnitude of curvatures of the decision boundary. Moosavi-Dezfooli et al. (2019) show the magnitude of the norm of gradients upper bound the largest eigenvalues of the hessian matrix of loss w.r.t. input $x$, thus measuring the curvature of the decision boundary. Besides, Moosavi-Dezfooli et al. (2019) even show that the low curvatures can lead to the enhanced robustness, which echoes our results in Figure 4.

The flatness values (red lines) increases abruptly at smaller learning rates (0.01, 0.001) at Epoch 30 and Epoch 60. It shows that when we begin to use adversarial data to fine-tune the decision boundary of the robust model, the decision boundary becomes more tortuous around the adversarial data (see Figure 9). This leads to the severe overfitting issue.

Similar to Figure 4, we compare GAIRAT and AT using the SVHN dataset, which can be found in Figure 10. Experiments on the SVHN dataset corroborate the reasons for issue of the robust overfitting and justify the efficacy of our GAIRAT. The training and evaluation settings keep the same as Figure 4 except the initial rate of 0.01 divided by 10 at Epoch 30 and 60 respectively.

## C.2 DIFFERENT LEARNING RATE SCHEDULES

In Figure 11, we compare our GAIRAT and AT using different learning rate schedules. Under the different learning rate schedules, our GAIRAT can relieve the undesirable issue of the robust overfitting, thus enhancing the adversarial robustness. To make the fair comparisons with Rice et al. (2020), we use the pre-activation ResNet-18 (He et al., 2016). We conduct standard adversarial training (AT) using SGD with 0.9 momentum for 200 epochs on CIFAR-10 dataset. The different learning rate schedules are in the top panel in Figure 11. The perturbation bound $\epsilon = 8/255$, the

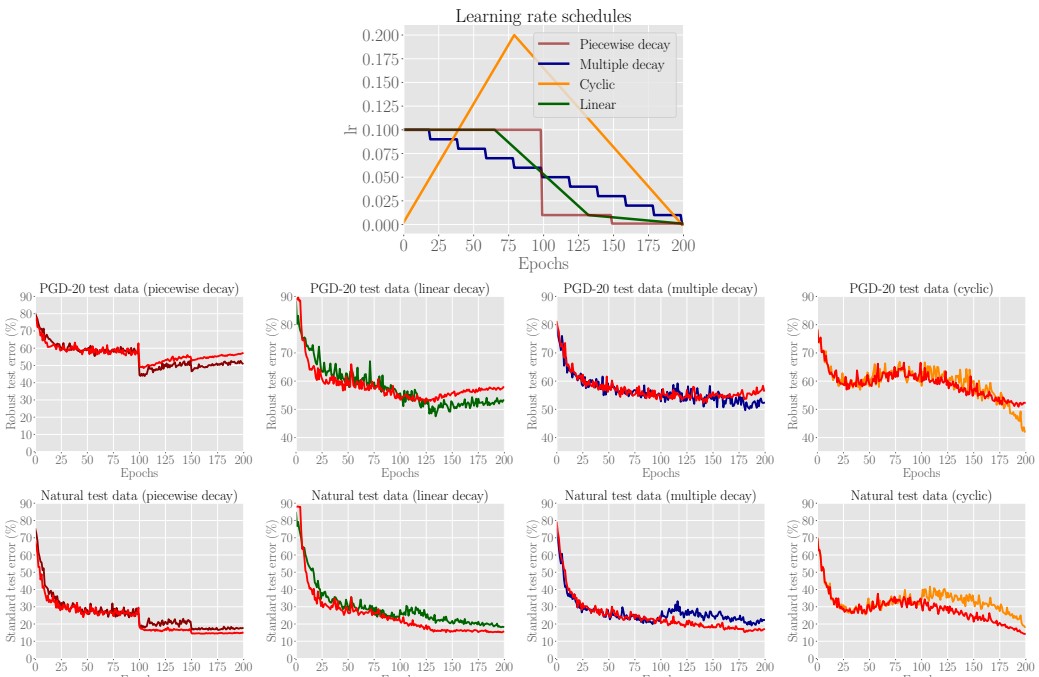

Figure 11: The training results of standard AT and GAIRAT using pre-activation ResNet-18 under different learning rate schedules on CIFAR-10 dataset. The top panel reports the different learning rate schedules. The four middle panels report the robust test error on adversarial data generated by PGD-20. The four bottom panels report the standard test error on natural data. The red lines represent AT's results under different learning rate schedules. The brown, green, blue and orange lines represent GAIRAT's results of different learning rate schedules.

PGD steps number $K = 10$ and the step size $\alpha = 2/255$. The training setting keeps the same as Rice et al. (2020)[1].

GAIRAT has the same training configurations (including all hyperparamter settings) including the 100 epochs burn-in period, after which, GAIRAT begins to introduce geometry-aware instance-reweighted loss. We use the weight assignment function $\omega$ from Eq. (6) with $\lambda = -1$.

At each training epoch, we evaluate each checkpoint using CIFAR-10 test data. In the middle panels of Figure 11, we report robust test error on the adversarial test data. The adversarial test data are generated by PGD-20 attack with the perturbation bound $\epsilon = 8/255$ and step size $\alpha = 2/255$. The PGD attack has a random start, i.e, the uniformly random perturbations of $[-\epsilon, \epsilon]$ are added to the natural data before PGD iterations, which keeps the same as Wang et al. (2019); Zhang et al. (2020b). Note that different from Rice et al. (2020) using PGD-10, we use PGD-20 because under the computational budget, PGD-20 is a more informative metric for the robustness evaluation. In the bottom panels of Figure 11, we report the standard test error on the natural data.

Figure 11 shows that under different learning rate schedules, our GAIRAT can relieve the issue of robust overfitting, thus enhancing the adversarial robustness with little degradation of accuracy.

## C.3 DIFFERENT WEIGHT ASSIGNMENT FUNCTIONS $\omega$

The weight assignment functions $\omega$ should be non-increasing w.r.t. the geometry value $\kappa$. In Figure 12, besides tanh-type Eq. (6) (blue line), we compare different types of weight assignment functions. The purple lines represent a linearly decreasing function, i.e.,

$$w(x, y) = 1 - \frac{\kappa(x, y)}{K + 1}. \tag{7}$$

---

[1]Robust Overfitting's GitHub

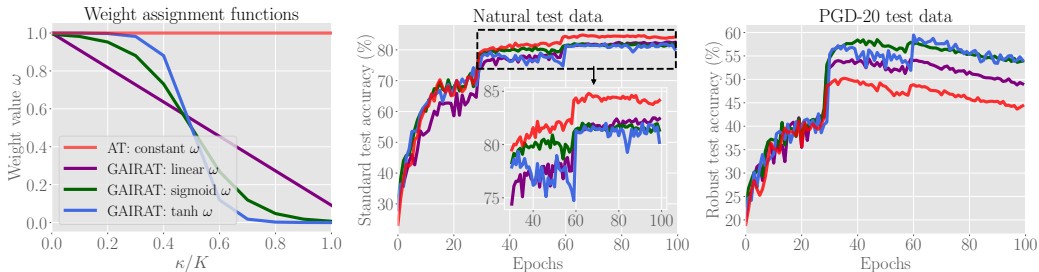

Figure 12: Comparisons of GAIRAT with different weight assignment functions on CIFAR-10 dataset. When GAIRAT takes constant $\omega = 1$ over the training epochs, GAIRAT recovers the standard adversarial training (AT) (red lines).

The green lines represent a sigmoid-type decreasing function, i.e.,

$$w(x, y) = \sigma(\lambda + 5 \times (1 - 2 \times \kappa(x, y)/K)), \tag{8}$$

where $\sigma(x) = \frac{1}{1+e^{-x}}$.

Figure 12 shows that compared with AT, GAIRAT with different weight assignment functions have similar degradation of standard test accuracy on natural data, but GAIRAT with the tanh-type decreasing function (Eq. (6)) has the better robustness accuracy. Thus, we further explore the Eq. (6) with different $\lambda$ in Figure 13.

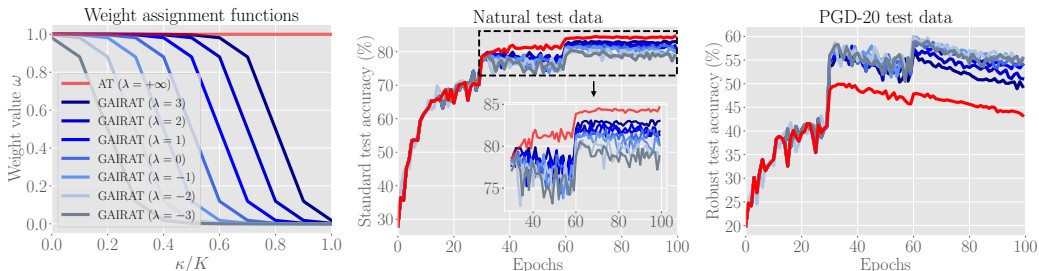

Figure 13: Comparisons of GARAT using the tanh-type weight assignment function (Eq. (6)) with different $\lambda$ on CIFAR-10 dataset.

In Figure 13, when $\lambda = +\infty$, GAIRAT recovers the standard AT, assigning equal weights to the losses of the adversarial data. Smaller $\lambda$ corresponds to the weight assignment function $\omega$, assigning relatively smaller weight to the loss of the adversarial data of the guarded data and assigning relatively larger weight to the loss of the adversarial data of the attackable data, which enhance the robustness more. With the same logic, larger $\lambda$ corresponds to the weight assignment function $\omega$, assigning relatively larger weight to the loss of the adversarial data of the guarded data and assigning relatively smaller weight to the loss of the adversarial data of the attackable data, which enhances the robustness less. The guarded data need more PGD steps $\kappa$ to fool the current model; the attackable data need less PGD steps $\kappa$ to fool the current model.

The results in Figure 13 justify the above logic. GAIRAT with smaller $\lambda$ (lighter blue lines) has better adversarial robustness with bigger degradation of standard test accuracy. On the other hand GAIRAT with larger $\lambda$ (darker blue lines) has relatively worse adversarial robustness with minor degradation of standard test accuracy. Nevertheless, our GAIRAT (light and dark lines) has better robustness than AT (red lines).

**Training and evaluation details** We training ResNet-18 using SGD with 0.9 momentum for 100 epochs. The initial learning rate is 0.1 divided by 10 at Epoch 30 and 60 respectively. The weight decay=0.0005. The perturbation bound $\epsilon = 0.031$; the PGD step size $\alpha = 0.007$, and PGD step numbers $K = 10$. For evaluations, we obtain standard test accuracy for natural test data and robust test accuracy for adversarial test data. The adversarial test data are generated by PGD-20 attack with

the same perturbation bound $\epsilon = 0.031$ and the step size $\alpha = 0.031/4$, which keeps the same as Wang et al. (2019). All PGD generation have a random start, i.e, the uniformly random perturbation of $[-\epsilon, \epsilon]$ added to the natural data before PGD iterations.

Note that the robustness reflected by PGD-20 test data is quite high. However, when we use other attacks such as C&W attack (Carlini & Wagner, 2017) for evaluation, both blue and red lines will degrade the robustness to around 40%. We believe this degradation is due to the mismatch between PGD-adversarial training and C&W attacks, which is the common deflect of the empirical defense (Tsuzuku et al., 2018; Wong & Kolter, 2018; Cohen et al., 2019; Balunovic & Vechev, 2020; Zhang et al., 2020a). We leave this for future work.

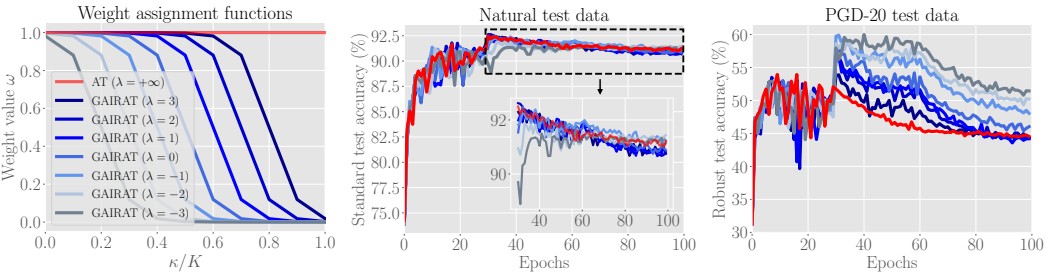

Figure 14: Comparisons of GARAT using the tanh-type weight assignment function (Eq. (6)) with different $\lambda$ on **SVHN** dataset.

In Figure 14, we also conduct experiments of GAIRAT using Eq. (6) with different $\lambda$ and AT using ResNet-18 on SVHN dataset. The training and evaluation settings keep the same as Figure 13 except the initial rate of 0.01 divided by 10 at Epoch 30 and 60 respectively.

Interestingly, AT (red lines) on SVHN dataset has not only the issue of robust overfitting, but also the issue of natural overfitting: The standard test accuracy has slight degradation over the training epochs. By contrast, our GAIRAT (blue lines) can relieve the undesirable robust overfitting, thus enhancing both robustness and accuracy.

## C.4 DIFFERENT LENGTHS OF BURN-IN PERIOD

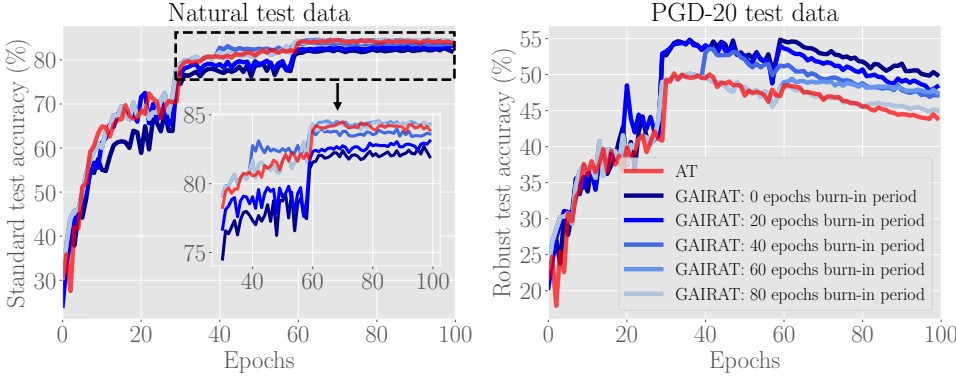

Figure 15: Comparisons of GAIRAT (blue lines) with different lengths of the burn-in period on CIFAR-10 dataset. The longer GAIRAT has the burn-in period, the more alike GAIRAT becomes standard AT (red lines). AT can be seen as GAIRAT with 100 epochs burn-in period. Darker blue lines represent shorter lengths of burn-in period; lighter blue lines represent longer lengths of burn-in period.

In Figure 15, we conduct experiments of GAIRAT under different lengths of the burn-in period using ResNet-18 on CIFAR-10 dataset. The training and evaluations details are the same as Appendix C.3 except the different lengths of burn-in period in the training.

Figure 15 shows that compared with AT (red lines), GAIRAT with a shorter length of burn-in period (darker blue lines) can significantly enhance robustness but suffers a little degradation of accuracy. On the other hand, GAIRAT with a longer length of burn-in period (lighter blue lines) slightly enhance robustness with zero degradation of accuracy.

## C.5    DIFFERENT NETWORKS - SMALL CNN AND VGG-13

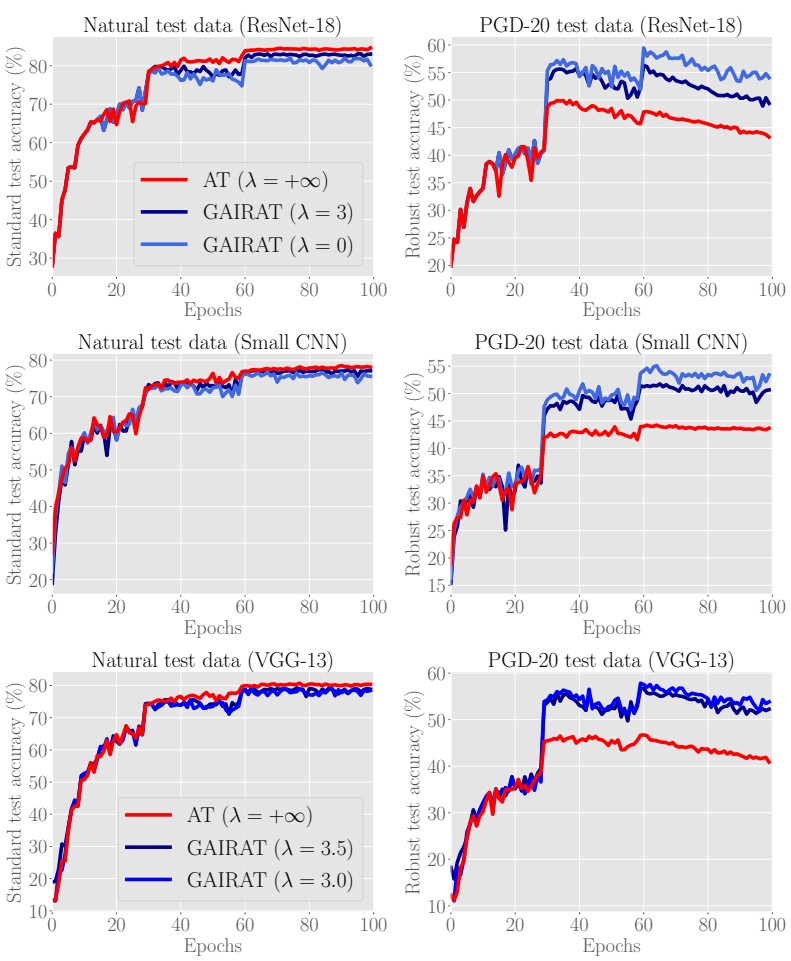

Figure 16: Comparisons of different networks (VGG-13, Small CNN and ResNet-18) which GAIRAT and AT use on CIFAR-10 dataset.

In Figure 16, besides ResNet-18, we apply our GAIRAT to Small CNN (6 convolutional layers and 2 fully-connected layers) on CIFAR-10 dataset. Training and evaluation settings keeps the same as the Appendix C.3; we use 30 epochs burn-in period and Eq. (6) as the weight assignment function.

Figure 16 shows that larger network ResNet-18 has better performance than Small CNN in terms of both robustness and accuracy. Interestingly, Small CNN has less severe issue of the robust overfitting. Nevertheless, our GAIRAT are still quite effective in relieving the robust overfitting and thus enhancing robustness in the smaller network.

In Figure 16, we also compare our GAIRAT with AT using VGG-13 (Simonyan & Zisserman, 2015) on CIFAR-10 dataset. Under the same training and evaluation settings as Small CNN, results of VGG-13 once again demonstrate the efficacy of our GAIRAT.

## C.6 GEOMETRY-AWARE INSTANCE DEPENDENT FAT (GAIR-FAT)

In this section, we show that GAIR-FAT can enhance friendly adversarial training (FAT). Our geometry-aware instance-reweighted method is a general method. Besides AT, we can easily modify friendly adversarial training (FAT) (Zhang et al., 2020b) to GAIR-FAT (See Algorithm 3 in the Appendix B.1).

In Figure 17, we compare FAT and GAIR-FAT using ResNet-18 on CIFAR-10 dataset. The training and evaluation settings keeps the same as Appendix C.3 except that GAIR-FAT and FAT has an extra hyperparameter $\tau$. In Figures 17, the $\tau$ begins from 0 and increases by 3 at Epoch 40 and 70 respectively. The burn-in period is 70 epochs. In Figure 17, we use Eq. (6) with different $\lambda$ as GAIR-FAT's weight assignment function.

Different from AT, FAT has slower progress in enhancing the adversarial robustness over the training epochs, so FAT can naturally resist undesirable robust overfitting. However, once the robust test accuracy reaches plateau, FAT still suffers a slight robust overfitting issue (red line in the right panel). By contrast, when we introduce our instance dependent loss from Epoch 70, GAIR-FAT (light and dark blue lines) can get further enhanced robustness with near-zero degradation of accuracy.

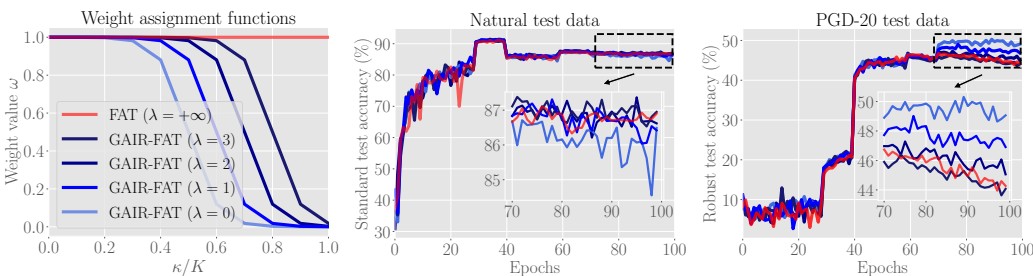

Figure 17: We compare FAT and GAIR-FAT using ResNet-18 on CIFAR-10 dataset using tanh-type weight assignment function with different $\lambda$.

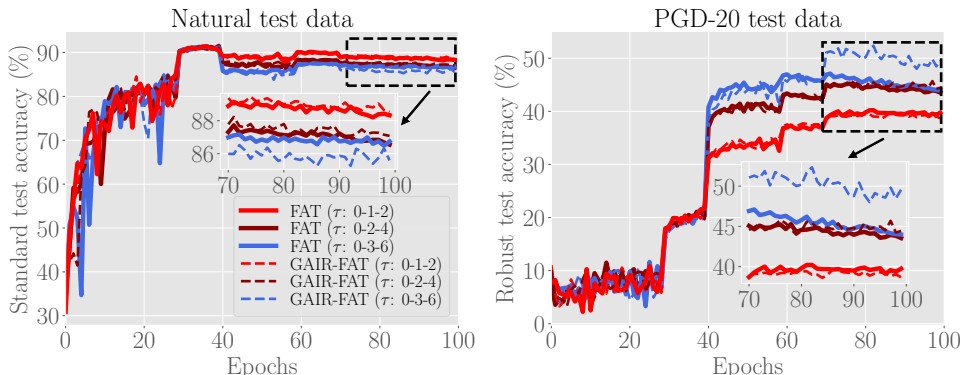

Figure 18: Comparisons of GAIR-FAT and FAT under different schedules of dynamical $\tau$ using ResNet-18 on CIFAR-10 dataset. ($\tau$: 0-1-2) refers to $\tau$ starting from 0 and increasing by 1 at Epoch 40 and 70, respectively. ($\tau$: 0-2-4) refers to $\tau$ starting from 0 and increasing by 2 at Epoch 40 and 70, respectively. $\tau$: 0-3-6) refers to $\tau$ starting from 0 and increasing by 3 at Epoch 40 and 70, respectively.

Note that different from the FAT used by Zhang et al. (2020b) increasing $\tau$ from 0 to 2 over the training epochs, we increase the $\tau$ from 0 to 6. As shown in Figure 18, we find out FAT with smaller $\tau$ (e.g., 1-3) does not suffer the issue of the robust overfitting, since the FAT with smaller $\tau$ has the slower progress in increasing the robustness over the training epochs. This slow progress leads to the slow increase of the portion of guarded data, which is less likely to overwhelm the learning from the attackable data. Thus, our geometry-aware instance dependent loss applied on FAT with smaller $\tau$ does not offer extra benefits, and it does not have damage as well.

## C.7 Geometry-aware Instance Dependent MART (GAIR-MART)

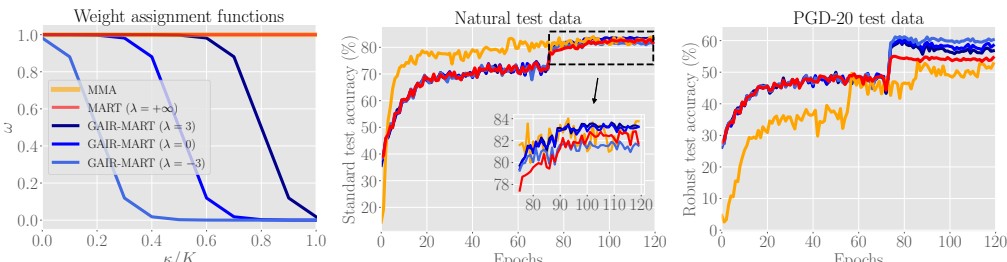

Figure 19: We compare MMA, MART and GAIR-MART with different weight assgginment functions using ResNet-18 on CIFAR-10.

In this section, we compare our method with MMA (Ding et al., 2020) and MART (Wang et al., 2020b). To be specific, we easily modify MART to a GAIRAT version, i.e., GAIR-MART. The learning objective of MART is Eq. (9); the learning objective of our GAIR-MART is Eq. (10).

The learning objective of MART is

$$\ell_{margin}(p(\tilde{x},\theta),y) + \beta\ell_{KL}(p(\tilde{x},\theta),p(x,\theta)) \cdot (1 - p_y(x,\theta)); \tag{9}$$

our learning objective of of GAIR-MART is

$$\ell_{GAIR_{margin}}(p(\tilde{x},\theta),y) + \beta\ell_{KL}(p(\tilde{x},\theta),p(x,\theta)) \cdot (1 - p_y(x,\theta)), \tag{10}$$

where $\ell_{margin} = -\log(p_y(\tilde{x},\theta)) - \log(1 - \max_{k\neq y} p_k(\tilde{x},\theta))$ and $p_k(x,\theta)$ is probability (softmax on logits) of $x$ belonging to class $k$. To be specific, the first term $-\log(p_y(\tilde{x},\theta))$ is commonly used CE loss and the second term $-\log(1 - \max_{k\neq y} p_k(\tilde{x},\theta))$ is a margin term used to improve the decision margin of the classifier. More detailed analysis about the learning objective can be found in (Wang et al., 2020b). In Eq. (9) and Eq. (10), $x$ is natural training data, $\tilde{x}$ is adversarial training data generated by CE loss, and $\beta > 0$ is a regularization parameter for MART. In Eq. (10), $\ell_{GAIR_{margin}} = -\log(p_y(\tilde{x},\theta)) \cdot \omega - \log(1 - \max_{k\neq y} p_k(\tilde{x},\theta))$ and $\omega$ refers to our weight assignment function.

For MMA and MART, the training settings keep the same as the [2] and [3]. For fair comparisons, GAIR-MART keeps the same training configurations as MART except that we use the weight assignment function $\omega$ (Eq.(6)) to introduce geometry-aware instance-reweighted loss from Epoch 75 onward. We train ResNet-18 on CIFAR-10 dataset for 120 epochs. For MMA, the learning rate is 0.3 from Iteration 0 to 20000, 0.09 from Iteration 20000 to 30000, 0.03 from Iteration 30000 to 40000, and 0.009 after Iteration 40000, where the Iteration refers to training with one mini-batch of data; For MART and GAIR-MART, the learning rate is 0.01 divided by 10 at Epoch 75, 90, and 100 respectively. For evaluations, we obtain standard test accuracy for natural test data and robust test accuracy for PGD-20 adversarial test data with the same settings as Appendix C.3.

Figure 19 shows GAIR-MART performs better than MART and MMA. The results demonstrate the efficacy of our GAIRAT method on improving robustness without the degradation of standard accuracy.

**Reweighing KL loss** The learning objective of MART explicitly assigns weights, not directly on the adversarial loss but KL divergence loss. We ask what if you replace their reweighting scheme $(1 - p_y(x,\theta))$ with our $\omega$. The learning objective is

$$\ell_{margin}(p(\tilde{x},\theta),y) + \beta\ell_{KL}(p(\tilde{x},\theta),p(x,\theta)) \cdot \omega. \tag{11}$$

Figure 20 reports the results: It does not have much effect on adding the geometry-aware instance-dependent weight to the regularization part, i.e., KL divergence loss .

---

[2]MMA's GitHub
[3]MART's GitHub

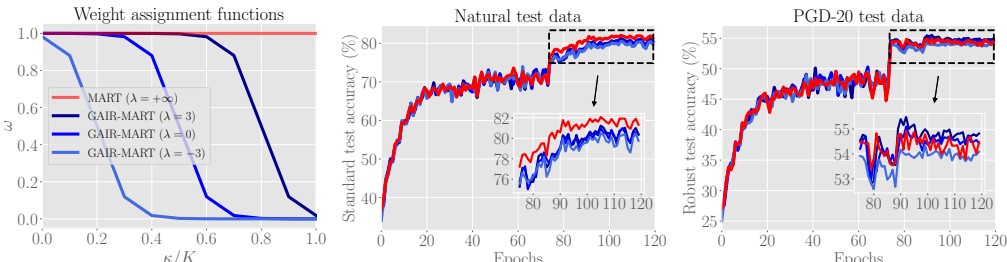

Figure 20: Comparison of MART and GAIR-MART training ResNet-18 with Eq. (11) on CIFAR-10 dataset.

### C.8    PERFORMANCE EVALUATION ON WIDE RESNET (WRN-32-10)

In Table 1, we compare our GAIRAT, GAIR-FAT with standard AT and FAT. CIFAR-10 dataset is normalized into [0,1]: Each pixel is scaled by 1/255. We perform the standard CIFAR-10 data augmentation: a random 4 pixel crop followed by a random horizontal flip. In AT, we train WRN-32-10 for 120 epochs using SGD with 0.9 momentum. The initial learning rate is 0.1 reduced to 0.01, 0.001 and 0.0005 at epoch 60, 90 and 110. The weight decay is 0.0002. For generating the adversarial data for updating the model, the perturbation bound $\epsilon_{\text{train}} = 0.031$, the PGD step is fixed to 10, and the step size is fixed to 0.007. The training settings come from FAT's Github. [4] In GAIRAT, we choose 60 epochs burn-in period and then use Eq. (6) with $\lambda = 0$ as the weight assignment function; the rest keeps the same as AT. The hyperparameter $\tau$ of FAT and begins from 0 and increases by 3 at Epoch 40 and 70 respectively; the rest keeps the same as AT. In GAIR-FAT, we choose 60 epochs burn-in period and then use Eq. (6) with $\lambda = 0$ as the weight assignment function; the rest keeps the same as FAT.

As suggested by results of the experiments in Section 4.1, the robust test accuracy usually gets significantly boosted when the learning rate is firstly reduced to 0.01. Thus, we save the model checkpoints at Epochs 59-100 for evaluations, among which, the best checkpoint is selected based on the PGD-20 attack since PGD+ is extremely computationally expensive. We also save the last checkpoint at Epoch 120 for evaluations. We run AT, FAT, GAIRAT and GAIR-FAT with 5 repeated times with different random seeds.

As for the evaluations, we test the checkpoint using three metrics: standard test accuracy on natural data (Natural), robust test accuracy on adversarial data generated by PGD-20 and PGD+. PGD-20 follows the same setting of the PGD-20 used by Wang et al. (2019)[5]. PGD+ is the same as $PG_{ours}$ used by Carmon et al. (2019)[6]. The adversarial attacks have the same perturbation bound $\epsilon_{test} = 0.031$. For PGD-20, the step number is 20, and the step size $\alpha = \epsilon_{test}/4$. There is a random start, i.e., uniformly random perturbations ($[-\epsilon_{test}, +\epsilon_{test}]$) added to natural data before PGD perturbations. For PGD+, the step number is 40, and the step size $\alpha = 0.01$. There are 5 random starts for each natural test data. Therefore, for each natural test data, we have $40 \times 5 = 200$ PGD iterations for the robustness evaluation.

In Table 1, the best checkpoint is chosen among the model checkpoints at Epochs 59-100 (selected based on the robust accuracy on PGD-20 test data). In practice, we can use a hold-out validation set to determine the best checkpoint, since (Rice et al., 2020) found the validation curve over epochs matches the test curves over epochs. The last checkpoint is the model checkpoint at Epoch 120. Our experiments find that GAIRAT reaches the best robustness at Epoch 90 (three trails) and 92 (two trails), and AT reaches the best robustness at Epoch 60 (five trails). FAT reaches the best robustness at Epoch 60 (four trails) and 61 (one trail). GAIR-FAT reaches the best robustness at around Epoch 90 (five trails). We report the median test accuracy and its standard deviation over 5 repeated trails.

**PGD attacks with different iterations**    In Table 1, each defense method has five trails with five different random seeds; therefore, each defense method has ten models (five last checkpoints and five best checkpoints). In Figure 21, for each defense, we randomly choose one last-checkpoint and

---

[4]FAT's GitHub

[5]DAT's GitHub

[6]RST's GitHub

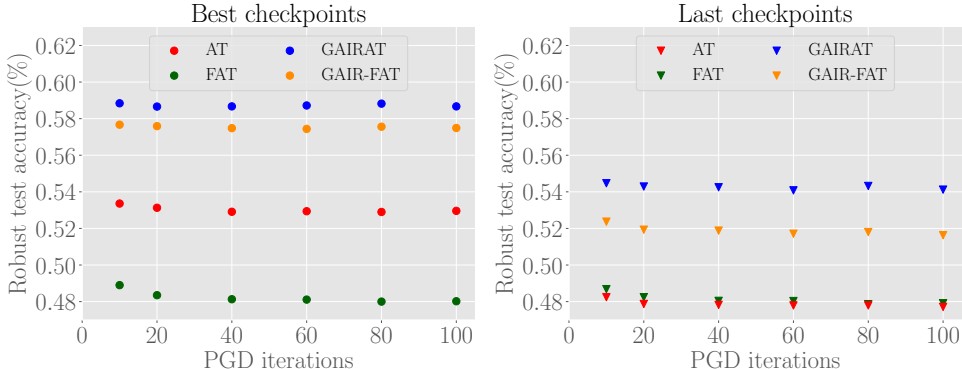

Figure 21: Comparison of PGD attacks with different PGD iterations on CIFAR-10 dataset.

one best-checkpoint and evaluate them using PGD-10, PGD-20, PGD-40, PGD-60, PGD-80, and PGD-100. All the PGD attacks use the same $\epsilon_{test} = 0.031$ and the step size $\alpha = (2.5 \cdot \epsilon_{test})/100$. We ensure that we can reach the boundary of the $\epsilon$-ball from any starting point within it and still allow for movement on the boundary, which is suggested by Madry et al. (2018). The results show the PGD attacks have converged with more iterations.

## C.9   PERFORMANCE EVALUATION ON WIDE RESNET (GAIR-TRADES)

Table 2: Test accuracy of TRADES and GAIR-TRADES (WRN-34-10) on CIFAR-10 dataset

| Defense | Best checkpoint | | | Last checkpoint | | |
|---|---|---|---|---|---|---|
| | Natural | PGD-20 | PGD+ | Natural | PGD-20 | PGD+ |
| TRADES ($\beta = 6$) | $84.88 \pm 0.35$ | $56.43 \pm 0.24$ | $54.33 \pm 0.38$ | $85.66 \pm 0.33$ | $53.31 \pm 0.25$ | $50.11 \pm 0.25$ |
| GAIR-TRADES ($\beta = 6$) | $\mathbf{86.99 \pm 0.31}$ | $\mathbf{63.32 \pm 0.50}$ | $\mathbf{56.77 \pm 0.87}$ | $\mathbf{86.86 \pm 0.26}$ | $\mathbf{60.65 \pm 1.00}$ | $\mathbf{52.70 \pm 0.93}$ |

In Table 2, we compare our GAIR-TRADES with TRADES. CIFAR-10 dataset normalization and augmentations keep the same as Appendix C.8. Instead, we use WRN-34-10, which keeps the same as Zhang et al. (2020b). We train WRN-34-10 for 100 epochs using SGD with 0.9 momentum. The initial learning rate is 0.1 reduced to 0.01 and 0.01 at epoch 75 and 90. The weight decay is 0.0002. For generating the adversarial data for updating the model, the perturbation bound $\epsilon_{train} = 0.031$, the PGD step is fixed to 10, and the step size is fixed to 0.007. Since TRADES has a trade-off parameter $\beta$, for fair comparison, our GAIR-TRADES uses the same $\beta = 6$. In GAIR-TRADES, we choose 75 epochs burn-in period and then use Eq. (6) with $\lambda = -1$ as the weight assignment function. We run TRADES and GAIR-TRADES five repeated trails with different random seeds.

The evaluations are the same as Appendix C.8 except the step size $\alpha = 0.003$ for PGD-20 attack, which keeps the same as Zhang et al. (2020b)[7].

In Table 2, the best checkpoint is chosen among the model checkpoints at Epochs 75-100 (w.r.t. the PGD-20 robustness). The last checkpoint is evaluated based on the model checkpoint at Epoch 100. Our experiments find that GAIR-TRADES reaches the best robustness at Epoch 90 (three trails), 96 (one trail) and 98 (one trail), and TRADES reaches the best robustness at Epoch 76 (three trail), 77 (one trails) and 79 (one trail). We report the median test accuracy and its standard deviation over 5 repeated trails.

Table 2 shows that our GAIR-TRADES can have both improved accuracy and robustness.

## C.10   BENCHMARKING ROBUSTNESS WITH ADDITIONAL UNLABELED (U) DATA

In this section, we verify the efficacy of our GAIRAT method by utilizing additional 500K U data pre-processed by Carmon et al. (2019) for CIFAR-10 dataset.

Carmon et al. (2019) scratched additional U data from 80 Million Tiny Images (Torralba et al., 2008); then, they used standard training to obtain a classifier to give pseudo labels to those U data.

---

[7]TRADES's GitHub

Among those U data, they selected 500K U data (with pseudo labels). Combining 50K labeled CIFAR-10's training data and pseudo-labeled 500K U data, they propose a robust training method named RST which utilized the learning objective function of TRADES, i.e.,

$$\ell_{CE}(f_\theta(x), y) + \beta\ell_{KL}(f_\theta(\tilde{x}), f_\theta(x)), \tag{12}$$

where $\tilde{x}$ is generated by PGD-10 attack with CE loss.

Based on the RST method, we introduce our instance-reweighting mechanism, i.e., our GAIR-RST. To be specific, we change the learning objective function to

$$\ell_{CE}(f_\theta(x), y) + \beta\left\{\omega\ell_{KL}(f_\theta(\tilde{x}), f_\theta(x)) + (1-\omega)\ell_{KL}(f_\theta(\tilde{x}_{CW}), f_\theta(x))\right\}, \tag{13}$$

where the $\tilde{x}_{CW}$ refers to the adversarial data generated by C&W attack (Carlini & Wagner, 2017) and $\omega$ is the as Eq. (6).

Table 3: Evaluations using standard WRN-28-10

| Method/Paper | Natural | AA |
|---|---|---|
| Gowal et al. (2020) | 89.48 | 62.60 |
| Wu et al. (2020) | 88.25 | 60.04 |
| **GAIR-RST (Ours)** | 89.36 | 59.64 |
| Carmon et al. (2019) | 89.69 | 59.53 |
| Sehwag et al. (2020) | 88.98 | 57.14 |
| Wang et al. (2020b) | 87.50 | 56.29 |
| Hendrycks et al. (2019) | 87.11 | 54.92 |

The results of other methods are reported at AA's GitHub

In Table 3, we compare the performance of our GAIR-RST with other methods that use WRN-28-10 under auto attacks (AA) (Croce & Hein, 2020). All the methods utilized the same set of U data which are from RST's GitHub[8] and the results are reported on the leaderboard of AA's GitHub[9]. Our GAIR-RST use the same training settings (e.g., learning rate schedule, $\epsilon_{train} = 0.031$) as RST. The evaluations are on the full set of the AA in (Croce & Hein, 2020) with $\epsilon_{test} = 0.031$, which keeps the same as training.

The results show our geometry-aware instance-reweighted method can facilitate a competitive model by utilizing additional U data.

---

[8]RST's GitHub
[9]AA's GitHub

