# OpenReview forum: "Geometry-aware Instance-reweighted Adversarial Training"
_ICLR.cc/2021/Conference — ICLR 2021 Oral_

### Official Review · AnonReviewer3 · 2020-10-27
**Paper332 AnonReviewer3**

**Rating:** 8
**Confidence:** 5

**Review:**

This paper challenges the common belief of the inherent tradeoff between robustness and accuracy.
Instead of recent methods improving accuracy while maintaining robustness, this paper proposes a geometry aware instance reweighed adversarial training (GAIRAT) method to improve robustness while maintaining accuracy.

Pros:
1 The direction---improving robustness while maintaining accuracy---is novel and interesting.

Specifically, several papers are challenging the inherent tradeoff, e.g., using more data [1], utilizing early stopped PGD [2], and incorporating dropout [3]. This paper still challenges the inherent tradeoff.
However, different from [2,3] improving accuracy while maintaining robustness, this paper goes the other direction.
To my knowledge, this is the first paper to explore this direction.

[1] Understanding and Mitigating the Tradeoff Between Robustness and Accuracy, ICML 2020
[2] Attacks Which Do Not Kill Training Make Adversarial Learning Stronger, ICML 2020
[3] A closer Look at Accuracy vs. Robustness, NeurIPS 2020

2 This paper has made two conceptual improvements. a) This paper explicitly argues that the overparameterized networks that have enough model capacity in standard training suffer from the insufficiency in adversarial training (though many studies have already shown AT needs the large model). b) This paper argues that under limited model capacity, adversarial data should have unequal importance. Unequal data's treatment was explored in the traditional ML methods several years ago, but it is rare in deep learning at this moment.

3 The proposed GAIRAT method is effective, indeed increasing robustness while retaining accuracy. The experiments are comprehensive over different network structures, datasets and attack methods. The experiments in the appendix provide much useful information.


Cons:
1.The design of weight assignment function in Section 3.3 seems heuristic. Would you explain some principles on assigning instance dependent weights?

2.In Figure 4, the GAIRAT method can relieve undesirable robust overfitting. Would you explain more about this? For example, why the robust overfitting exists in standard adversarial training? how/why your GAIRAT methods relieve it?

---

> ### Author Response · Authors · 2020-11-19
> **Replies to Reviewer 3**
>
> Many thanks for the great comments. Please find our replies below.
>
> 1 Explain some principles on designing weight assignment functions.
>
> The optimal weight assignment is still an open question. Therefore, we have conduct experiments in Section C.3 and C.4, empirically evaluating different weight assignment functions $\omega$ and different starting epochs to apply $\omega$. \
> In terms of some principles on designing the weight assignment function $\omega$,\
> (a) In GAIRAT, the weight assignment is non-increasing w.r.t. the $\kappa$ value of the natural data. It reflects different degrees of focus on different data. \
> GAIRAT puts more focus on attackable data whose adversarial variants are easily misclassified and less focus on guarded data whose adversarial variants are hardly misclassified.\
> (b) The design of $\omega$ should be dataset-aware. For example, a suitable $\omega$ applies to the CIFAR-10 dataset may not perfectly fit the SVHN dataset. \
> Compared with the CIFAR-10 dataset (in Figure 4), the portion of guarded SVHN data (in Figure 10) becomes very large ($\kappa = 10$) at very initial training epochs. A better $\omega$ that is aware of this phenomenon may further increase the performance. \
> (c) The design of $\omega$ should be aware of the training stage. At different epochs, learning may need different $\omega$ for the reweighting instance-dependent adversarial losses. \
> We leave these explorations for future work.
>
> 2 More explanations on robust overfitting.
>
> In Section C.1, we have extensive discussions and more experiments on **GAIRAT relieve robust overfitting**.\
> -why the robust overfitting exists in standard adversarial training? \
> As the training progresses, the adversarially robust model engenders the increasing number of guarded training data (larger $\kappa$ value in bottom left panel in Figure 1) and the decreasing number of attackable training data (smaller $\kappa$ value).
> Equally focusing on training on the adversarial data may cause a vast number of adversarial variants of guarded data to *overwhelm* the model during the training, leading to undesirable robust overfitting.\
> -how/why your GAIRAT methods relieve it?\
> GAIRAT explicitly assigns less weight on the large portion of guarded data and assigns more weight on the small portion of attackable data, therefore ameliorating this *overwhelm* issue. \
> As a result, our GAIRAT can facilitate a flatter loss landscape. This fact is manifested in the bottom-middle and -right panels in Figure 4 and Figure 10; the illustrations are in Figure 9.

---

### Official Review · AnonReviewer4 · 2020-10-27
**Reviews for GAIRAT**

**Rating:** 8
**Confidence:** 5

**Review:**

This paper focuses on adversarial learning. It improves the robustness while keeping the accuracy. To achieve this point, the authors find that adversarial data should have unequal importance, which naturally brings geometry-award instance-reweighted adversarial training (GAIRAT).

Pros:
1. The paper has strong novelty in philosophy level. The common belief is that robustness and accuracy hurt each other. However, this paper shows that the robustness can be improved while keeping accuracy. As far as I know, this point has never been explored before.

2. The paper is well motivated and easy to follow. First, the authors use Figure 1 to illustrate the GAIRAT, which explicitly gives larger weights on the losses of adversarial data. The authors use two toy examples in Figure 3 to explain GAIRAT more. Second, the whole logic of this paper is easy to follow. For example, after explaining motivations of GAIRAT, we can clearly see the objective function of GAIRAT and its realization.

3. The paper is sufficiently justified in experiments. For example, PGD-200 has been used to verify the robustness of GAIRAT. From my personal opinion, this result is quite strong. Moreover, the authors upgrade their method by incorporing FAT and verify the robustness of GAIR-FAT.

Cons:
1.In the top right panel of Figure 10, the SVHN experiments have a period of increasing robustness training error for GAIRAT. Could you explain this?

2.Although authors show that model capacity is not enough in adversarial training, how large the DNN should be enough? What do you think?

---

> ### Author Response · Authors · 2020-11-19
> **Replies to Reviewer 4**
>
> Many thanks for the great comments. Please find our replies below.
>
> 1 SVHN experiments have a period of increasing robustness training error in Figure 10.
>
> In the SVHN dataset, we believe this is due to the shortage of adversarial training data at the later training stage. \
> As the training progresses, most natural data quickly reach the $\kappa$ (the number of PGD steps needed to fool the current model) value up to 10 (red lines in the bottom left panel). \
> Our weight assignment function (in the top left panel) assigns zero weights to the losses of adversarial data whose natural data have $\kappa = 10$; thus, very few adversarial data are utilized to update the model at the later training stage. \
> When trained with very few data points, the robust training error gets increased.
>
> 2 How large the DNN should be enough for adversarial training?
>
> This is still an open question, which is very interesting. Although there exist no exact answers, I can still provide some insights. \
> (a) Slightly larger defense parameter $\epsilon_{train}$ usually requires significantly larger models.
> Adversarial training forces DNN to memorize the natural data's local neighborhood; this local neigborhood is exponentially large w.r.t. input dimensions, i.e. $||1+\epsilon_{train}||^{input \ dim}$. \
> Therefore, even a slightly larger $\epsilon_{train}$ can significantly enlarge the local neighborhood. Smoothing the large neighborhood requires larger models. \
> (b) From this neighborhood smoothing perspective, I guess the current network structure, e.g., (Wide) ResNets, may not cater to the input smoothing. \
> For example, when networks become very deep or wide, the amount of tunable parameters is tremendous, which not only makes the decision boundary very complicated but also hurdles the optimization.\
> Therefore, a new type of network structure catering to local smoothing (adversarial training) is encouraged, rather than purely focusing on increasing the network size. \
> (c) Optimization is difficult in adversarial training. \
> For example, Zhang et al. (2020) showed adversarial training has cross-over mixture issues, which can potentially *kill* the learning [1].\
> Therefore, a new optimization caters to adversarial training is encouraged. \
> [1] Attacks Which Do Not Kill Training Make Adversarial Learning Stronger, ICML 2020

---

> > ### Comment · AnonReviewer4 · 2020-11-23
> > **Thanks for the rebuttal**
> >
> > I read through the responses and other reviewers' comments. The authors have addressed my questions and I support acceptance.

---

### Official Review · AnonReviewer1 · 2020-10-27
**Interesting paper**

**Rating:** 7
**Confidence:** 4

**Review:**

Summary:
The paper focused on the sample importance in the adversarial training. The authors firstly revealed that over-parameterized deep models on natural data may have insufficient model capacity for adversarial data, because the training loss is hard to zero for adversarial training. Then, the authors argued that limited capacity should be used for these important samples, that is, we should not treat samples equally important. They used the distance to the decision boundary to distinguish important samples and proposed geometry-aware instance-reweighted adversarial training. Experiments show the superiority over baselines.

Pros:
- The finding on insufficient model capacity is very interesting. The following motivation for GAIRAT is intuitive and well explained.
- The authors proposed a realized measurement to compute the distance to the decision boundary. This is inspiring for a series of decision-based work.
- The experiments demonstrate the effectiveness of the proposed method.

Cons:
- Treating data differently has been investigated in related work like MART and MMA. The authors should discuss the difference from these methods.
- The capacity analysis provides a very good perspective to analyze adversarial training, however, the explanations in Figure 2 are a little bit weak.
- The weight function of Eq. (6) lack some intuitive explanations. Why such a formula? Why choose these constants?
- PGD steps are also investigated in CAT and DAT papers. The authors should also discuss the difference to them.
- The experiments should compare with some baselines considering the example difference, such as MART, MMA.
- The evaluations should test some modern white-box attacks, like auto-attack, only PGD is not convincing. Besides, Black-box attacks should be tested for a complete evaluation and checking the obfuscated gradients.

---

> ### Author Response · Authors · 2020-11-19
> **Replies to Reviewer 1**
>
> Many thanks for the great comments! Please find our replies below.
>
> 1 Discuss the difference with related work. (MART, MMA, CAT, and DAT)
>
> MMA, CAT, and DAT generated differently adversarial data for updating the model. Specifically, the adversary strength is measured by PGD steps (CAT), convergence quality (DAT), and perturbations bound epsilon (MMA). \
> Different from those existing methods, our GAIRAT treats adversarial data differently by explicitly assigning different weights on the adversarial loss of adversarial data. Explicitly assigning weights has the benefit of breaking the ``blocking effect’’ (The blocking effect is stated in Section 3.2). \
> Note that MART's learning objective also explicitly assigned weights, not directly on the adversarial loss but KL divergence loss. The KL divergence loss helps to strengthen the smoothness within the norm ball of natural data, which is also used in VAT and TRADES. \
> Differently from MART, our GAIRAT explicitly assigns weights on the adversarial loss. Therefore, we can easily modify MART to GAIR-MART. \
> Besides, MART assigns weights based on the model’s prediction confidence on the natural data. GAIRAT assigns weights based on how easy the natural data can be attacked. \
> We have updated the main paper in Section 3.3 adding those discussions.
>
> 2 Compare experiments with MART and MMA.
>
> We have updated Appendix C.7 comparing MMA, MART, and our GAIR-MART. \
> The experiments show our GAIRAT outperforms the baselines.
>
> 3 Weak explanations in Figure 2.
>
> We have updated Figure 2 by adding the learning curve of standard training (red line). \
> There is a big gap between the red line and blue lines. \
> The over-parameterized networks that can easily memorize all data in standard training, find it difficult to fit data (both natural data and adversarial data) in adversarial training.\
> Could I know in which part I can strengthen the explanations?
>
> 4 The weight function of Eq (6) lacks some intuitive explanations.
>
> In GAIRAT, weight assignment function $\omega$ is non-increasing w.r.t. the geometry value $\kappa$. \
> Eq. (6) is just one example, which is fungible. In Section C.3, we have discussed different types of $\omega$. Experiments show all non-increasing $\omega$ can enhance robustness significantly. \
> For more intuitive explanations,  $\omega$ serves for the purpose of enforcing the different focus by the optimizer. The optimizer will focus less on already-guarded data and focus more on those attackable data. \
> The choices of formula and the constants are hyperparameters dependent on various datasets & learning tasks. \
> It is still an open question on choosing the optimal weight assignment functions; we will leave this as future work.
>
> 5 Evaluations using AA attacks.
>
> We have leveraged 500K unlabeled data (preprocessed by Carmon et al. 2019). Our geometry-aware instance-reweighed method can still facilitate a good WRN-28-10 model in terms of both robustness and accuracy. \
> We evaluate the model using auto attack (AA). AA attack is a combination of two white-box attacks and two black-box attacks.
> The standard test accuracy is 89.36%, and AA attack accuracy (on the full test set) is 59.64%. \
> We have added Appendix C10 illustrating the details and the results. \
> For the code, you can check the updated attachment for the training details and verifying our methods.

---

### Public Comment · ~Nicholas_Carlini1 · 2020-11-13
**How does the defense perform with more iterations of PGD?**

This paper evaluates against 40 iterations of PGD, where the strongest result is then +7% accuracy. By repeating this to 5 random restarts, the increase goes down to +4.2% accuracy. What happens if you run for more (several hundred) iterations of gradient descent? Does the effect size continue to decrease?

Note there is a difference between N iterations of gradient descent repeated M times, and N*M iterations of gradient descent. It is not always clear how to perform this tradeoff, but in many cases in the past 40 iterations of PGD has not been sufficient to converge.

For example, it may help to introduce a plot that shows accuracy as a function of PGD steps, as done for example in Figure 1 of Madry et al. 2017. Note that here the stop at 100 because the attack has (mostly) converged---you may have to try more if things have not converged by 100 iterations.

---

> ### Author Response · Authors · 2020-11-16
> **PGD+ is strong enough and converged.**
>
> Many thanks for your question!
>
> From your question, I guess you are looking at AT and GAIRAT’s comparisons at the last-checkpoint in Table 1.
> Nevertheless, We have conduct PGD-200 for evaluations on models in Table 1. The results---median (std)---are
>
> Defense (best checkpoint)| PGD-200 Acc. | PGD+ Acc. | \
> |AT(Madry) |51.76 (0.23) | 51.28 (0.23) |\
> | FAT |46.63 (0.18) | 46.14 (0.19) |\
> | GAIRAT |57.81 (0.49) | 55.61 (0.61)|\
> | GAIR-FAT |56.27 (0.53) | 53.50 (0.60) |
>
>  Defense(last checkpoint)| PGD-200 Acc. | PGD+ Acc. | \
> |AT(Madry) | 46.46 (0.05) | 46.08 (0.07) |\
> | FAT | 46.36 (0.24) | 45.80 (0.16) |\
> | GAIRAT |53.61 (0.49) | 50.32 (0.48) |\
> | GAIR-FAT | 50.36 (0.55) | 47.51(0.51) |
>
> PGD+ is PGD with five random starts, and each start has 40 steps with step size 0.01 (PGD+ has 40 × 5 = 200 iterations for each test data).  PGD-200 is one random start with 200 steps with step size 0.001.
> *$\epsilon_{test}$ = 0.031, which is very small, where adversarial data cannot exceed the small norm ball; compared with 0.01, step size of 0.001 is more fine-tuned for PGD-200. Therefore, PGD-200 have converged.
>
> From the above table, PGD + is stronger than PGD-200 because PGD is easily trapped in local minima.
>
> Besides, we updated Appendix C.8 adding a plot that shows accuracy as a function of PGD steps. We have found that the attacks are converged on GAIRAT models.

---

### Comment · ~Ming_Xiao1 · 2021-03-03
**An interesting paper on today's arXiv**

This paper easily breaks GAIRAT by a simple logic scaling attack

https://arxiv.org/abs/2103.01914

---

> ### Comment · ~Jingfeng_Zhang1 · 2021-03-03
> **Many thanks for bring us this very interesting paper.**
>
> Dear Ming Xiao,
>
> Many thanks for bring us this interesting paper.
>
> For the similar matter, you can also refer to this GitHub link <https://github.com/zjfheart/Geometry-aware-Instance-reweighted-Adversarial-Training/issues/1>.
>
> GAIRAT training under the PGD attacks can defend PGD attacks very well, but indeed, it cannot perform equally well on all existing attacks.
> From the philosophical perspective, we cannot expect defense under one specific attack can defend all existing attacks. This philosophy echoes the previous finding that "in adversarial training, it is essential to include adversarial examples produced by all known attacks, as the defensive training is non-adaptive. [1]"
>
> One potential solution is to incorporate all existing attacks into the training procedure.
> GAIRAT actually is not quite a specific method, it is an idea, i.e., data are inherently different, and adversarial training should treat data differently.
> Incorporating all attacks in GAIRAT yet preserving the efficiency is an interesting future direction.
>
> Besides, you can refer to Appendix C.10 to check the AA attack results.
>
> Again, many thanks for bring us attention the paper.
>
> [1] SoK: Towards the Science of Security and Privacy in Machine Learning.
>
> Best wishes,\
> Jingfeng

---

> > ### Comment · ~Ming_Xiao1 · 2021-03-03
> > **Logic Scaling Attack is simple, basic and as important as PGD**
> >
> > Thanks for your response.
> >
> > I would say logic scaling attack is as simple, basic, and important as PGD. Anyone can make any NN near 100% robust under vanilla PGD by only changing the temperature (e.g. 0.001) of cross-entropy loss. And it is a long-lasting pitfall when evaluating the robustness. It's hard to believe that the community still knows little about the logic scaling attack in 2021 (five years after it was originally proposed).
> >
> > See the following paper
> > https://arxiv.org/pdf/1607.04311.pdf
> >
> > @Nicholas Carlini

---

> > ### Comment · ~Iacopo_Masi2 · 2021-03-05
> > **A few more comments on the Logit Scaling Attack**
> >
> > Dear All,
> >
> > we are the authors of the report here. Thanks to Ming to bring our report up and thanks to Jingfeng for his answer. First of all, we believe GAIRAT is a valuable method and the rationale behind makes sense. I came up with something similar but then after some time, I discovered at my own expense that was subject to gradient masking technique related to scaling the logits. When I read GAIRAT, it was interesting and I had the hunch that it could be prone to this attack and we tested it out and wrote the short report.
> >
> > We were not aware the logit scaling attack was shown firstly by Carlini [1], our attempt was inspired by Section 4 of the AA paper [2]. Thank you, Ming, for bringing this to our attention.
> >
> > In light of this, we are going to update our report by citing [1] as well. We also are aware only now by looking at the GitHub issues that the follow-up paper [3] shows the same result; despite that, we believe is important to remind **why** the accuracy goes down (and not just show that it does) related to the gradient masking produced by defenses that scale the baseline CE loss. We hope that this can raise awareness in our community.
> >
> > Best Regards,
> >
> > [1] https://arxiv.org/pdf/1607.04311.pdf
> >
> > [2] https://arxiv.org/pdf/2003.01690.pdf
> >
> > [3] https://arxiv.org/pdf/2102.07327.pdf

---

### Comment · ~Gang_Niu1 · 2021-03-06
**Reply to "A few more comments on the Logit Scaling Attack" by Iacopo Masi**

Thanks for your attention to our work! I hereby discuss some issues on your tech report and the logit attack itself one by one.

A1. First of all, the importance weighting (IW) implementation we have released on github is just one of possible implementations of GAIRAT. There are many other ways to map the number of PGD steps to instance weights, and the number of PGD steps is only a rough approximation of the geometry property we would like to model, i.e., the non-linear distance from an instance to the class boundary it belongs to. Unlike IW for distribution shifts where we know the optimal weights, we have no idea that which IW map is the best, at least theoretically.

A2. Hence, the IW map itself may be a hyperparameter depending on the task --- the data, the loss, the model, the optimizer, and the set of future attacks. Technically, if we consider flatter IW maps, GAIRAT becomes closer to the standard AT; indeed, GAIRAT with possible IW maps can be regarded as a strictly general case of the standard AT since GAIRAT is just AT equipped with IW.

A3. Moreover, even if the IW map is fixed rather than tuned, other hyperparameters such as weight decay, learning rate, and the choice of the "best epoch" should be tuned on validation data, where the validation data are attacked by some stronger test-time attack instead of the weaker training-time attack. Note that AT is much more sensitive to changes of hyperparameters than natural training without simulating the attack; see "Bag of Tricks for Adversarial Training" (ICLR 2021: https://openreview.net/forum?id=Xb8xvrtB8Ce).

Then, let me talk about the logit attack itself.

B1. The community knew the attack long time ago and knew how to cancel its effect by actively scaling the weights and biases in the layer where soft-max serves as the activation. If the logit values will be multiplied by 10, the weights and biases will be divided by 10. Then the attacker can argue that the logit values will be multiplied by 100, and so on and so forth... This is an endless game. Even though the attacker can multiply the logit values by a very large number like 1 million, as long as the defender first divide the weights and biases by a much larger number like 1 billion, the robust accuracy can be as close to the natural accuracy as possible (not close to 100%). Therefore, in recent years, almost no paper has played this endless game, since the protocol of experimenting AT algorithms is that neither attacker nor defender is allowed to actively scale the logit values.

B2. Furthermore, the central philosophy of AT is to simulate the future attacks. Certainly, we would like to consider the diversity of attacks, but if we are only allowed to simulate a single attack, we will simulate the strongest attack that is still computationally affordable. By "computationally affordable", I meant we cannot simulate PGD-100 even when we know that the attacker will use PGD-100, but we can simulate PGD-10 with eps=16/255 when we know that the attacker will use either PGD-10 or PGD-100 with eps=16/255 rather than eps=8/255 on CIFAR-10. Similarly, multiplying the logit values by 10 is computationally affordable, and if doing so makes the attack stronger than before, we can simulate and should simulate the stronger attack during training.

B3. In my humble opinion, science and engineering are quite different. Kaggle newcomers know that models should be ensembled in practice, but almost no NeurIPS, ICML, or ICLR paper did that unless the paper works on ensemble learning itself. This is science vs engineering. In science, we don't consider all factors for the ease of ablation study. We cannot conclude that it is hard to believe top ML researchers still don't know ensemble learning after it has been proposed for 40+ years. There is significant difference between knowing a thing, knowing how to properly use a thing, and knowing when/why should/shouldn't use a thing.

Thanks again for your attention to our work, and hope my explanations above clarified some of your concerns.

---

> ### Comment · ~Gang_Niu1 · 2021-03-06
> **The formulation of AT is more general than the formulation of the logit attack**
>
> There is another reason why the logit attack is not popularly considered. It requires that the surrogate loss function is the soft-max cross entropy. It works at most for proper composite losses [1,2], where the scores (cf. logit values) are transformed by a link function (strictly speaking, an inverse link function in probability theory; cf. soft-max) and then the estimated class-posterior probabilities are sent to a base loss (cf. cross entropy). It cannot work if the surrogate loss is directly computed based on the scores, including but not limited to the multi-class margin loss, the one-vs-rest loss, and the pairwise comparison loss [3--8]. Without the transformation by a link function, there is no effect of gradient masking.
>
> Nevertheless, in the formulation of AT, the surrogate loss can be any loss, not necessarily probabilistic, let alone proper composite.
>
> [1] Mark D. Reid and Robert C. Williamson. Composite binary losses. JMLR, 11:2387--2422, 2010.
>
> [2] Robert C. Williamson, Elodie Vernet, and Mark D. Reid. Composite multiclass losses. JMLR, 17:1-52, 2016.
>
> [3] Koby Crammer and Yoram Singer. On the algorithmic implementation of kernel-based vector machines. JMLR, 2:265--292, 2001.
>
> [4] Jason Weston and Chris Watkins. Multi-class support vector machines. Technical Report CSDTR-98-04, Department of Computer Science, Royal Holloway College, University of London, 1998.
>
> [5] Yoonkyung Lee, Yi Lin, and Grace Wahba. Multicategory support vector machines: Theory and application to the classification of microarray data and satellite radiance data. Journal of the American Statistical Association, 99(465):67--81, 2004.
>
> [6] Tong Zhang. An infinity-sample theory for multi-category large margin classification. NeurIPS, 2004a.
>
> [7] Tong Zhang. Statistical analysis of some multi-category large margin classification methods. JMLR, 5:1225--1251, 2004b.
>
> [8] Tong Zhang. Statistical behavior and consistency of classification methods based on convex risk minimization. Annals of Statistics, 32(1):56--85, 2004c.

---

> ### Comment · ~Gang_Niu1 · 2021-03-06
> **My final note on the logit scaling trick**
>
> I would like to deliver my final note concerning Figure 1 in the tech report entitled "Evaluating the Robustness of Geometry-Aware Instance-Reweighted Adversarial Training".
>
> While dividing the logit values by a large number can make PGD attacks weaker, multiplying by a large number does not necessarily make PGD attacks stronger. Of course, if the soft-max outputs are flatter, the signal-to-noise ratio of the following gradient-based attack algorithms goes lower, and if the soft-max outputs are sharper, the signal-to-noise ratio goes higher and it seems more friendly to the following gradient-based attack algorithms. However, both of these operations introduce an additional bias so that the observed gradients are no longer the loss-maximizing directions. Therefore, there exists a bias-variance tradeoff when we want to make PGD attacks stronger by scaling the logit values, but there is no tradeoff when we want to make PGD attacks weaker by scaling the weights and biases before the soft-max activation.
>
> This explanation is consistent with Figure 1 in the tech report.

---

### Decision · Program_Chairs · 2021-01-07
**Final Decision**

**Decision:**

Accept (Oral)

**Comment:**

The paper proposes an insightful study on the robustness and accuracy of the model. It was hard to simultaneously keep the robustness and accuracy. A few works tried to improve accuracy while maintaining the robustness by investigating more data, early stopping or dropout. From a different perspective, this paper aims to improve robustness while maintaining accuracy.

There are some interesting findings in this paper, which could deepen our understanding of adversarial training. For example, the authors conducted experiments with different sizes of the network in standard training and adversarial training. The capacity of an overparameterized network can be sufficient for standard training, but it may be far from enough to fit adversarial data, because of the smoothing effect. Hence given the limited model capacity, adversarial data all have unequal importance. Though this technique is simple and widely studied in traditional ML, it is an interesting attempt in adversarial ML and the authors provide extensive experimental results to justify its effectiveness.

In the authors' responses, the concerns raised by the reviewers have been well addressed. The new version becomes more complete by including more results on different PGD steps and the insights on designing weight assignment function. Also, the authors gave an interesting discussion on enough model size for the adversarial training, though it is still kind of an open question. I would thus like to recommend the acceptance of this paper.